# INCODE: INTERPRETABLE COMPRESSED DESCRIPTIONS FOR IMAGE GENERATION

**Armand Comas[1]**   **Aditya Chattopadhyay[2]***   **Feliu Formosa[1]**   **Changyu Liu[1]**
**Octavia Camps[1]**   **René Vidal[3]**
[1]Northeastern University   [2]Johns Hopkins University   [3]University of Pennsylvania
{comasmassague.a, formosa.f, liu.changyu, o.camps}@northeastern.edu
achatto1@jhu.edu
vidalr@seas.upenn.edu

## ABSTRACT

Generative models have been successfully applied in diverse domains, from natural language processing to image synthesis. However, despite this success, a key challenge that remains is the ability to control the semantic content of the scene being generated. We argue that adequate control of the generation process requires a data representation that allows users to access and efficiently manipulate the semantic factors shaping the data distribution. This work advocates for the adoption of succinct, informative, and interpretable representations, quantified using information-theoretic principles. Through extensive experiments, we demonstrate the efficacy of our proposed framework both qualitatively and quantitatively. Our work contributes to the ongoing quest to enhance both controllability and interpretability in the generation process. Code available at github.com/ArmandCom/InCoDe.

## 1 INTRODUCTION

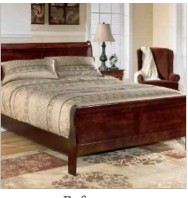
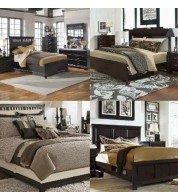
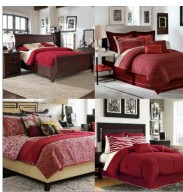
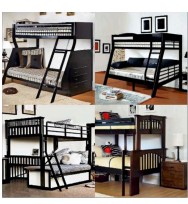

Reference (a)     Re-generation (b)

Are the bedsheets red? No →Yes     Does it have bunkbeds? No →Yes
Does it have a window? Yes →No

(c)

Figure 1: `InCoDe` Controllable Image Generation and Editing: a user can provide a sample image (a) and request to generate new images that are semantically "similar" to the given one (b). In addition, the user can also encode their preferences in the generated images by specifying certain attributes like color of bedsheets (c).

The ability to control the output of a generative model is essential for practical applications in various domains, ranging from natural language processing to image synthesis. For example, consider a scenario in which users may be interested in using an online application to decorate their bedroom. This application could generate images that are semantically similar to an "inspiration" picture given by users, as depicted in Fig.1(a, b), while allowing them to specify which aspects they want to preserve or change (Fig. 1(c)). Alternatively, if the user does not provide an image, the application could ask them a series of questions about their preferences and generate a set of image suggestions based on their answers, as seen in Fig. 2(d). To speed up this process and maximize user satisfaction, the app should prioritize asking the most relevant questions upfront, reducing the number of interactions while ensuring the user's preferences are accurately captured.

Designing such an application requires an image representation that effectively captures the elements and properties most semantically significant to the user for the specific task (e.g., bedroom decoration). The representation should be interpretable and easily modifiable based on the user's input. In addition, it should allow for measuring relevance and similarity within its domain. Using such a representation, the application could offer personalized suggestions and allow precise adjustments to the generated images, ultimately improving the user experience.

---

*Work done prior to joining Amazon.

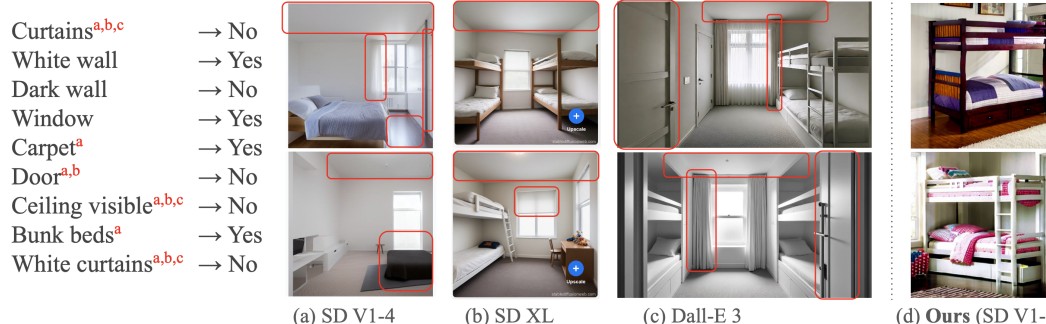

| Curtains[a,b,c] | → No |
| White wall | → Yes |
| Dark wall | → No |
| Window | → Yes |
| Carpet[a] | → Yes |
| Door[a,b] | → No |
| Ceiling visible[a,b,c] | → No |
| Bunk beds[a] | → Yes |
| White curtains[a,b,c] | → No |

(a) SD V1-4  (b) SD XL  (c) Dall-E 3  (d) **Ours** (SD V1-4)

Figure 2: **Text-to-image models fail when composing multiple concepts**. Comparison with text-to-image baselines. For a given subset of attributes: (a) Samples from Stable Diffusion v1-4 ($\approx \times 0.95$ params w.r.t. `InCoDe`). (b) Samples from Stable Diffusion XL ($\approx \times 10$ params). (c) Samples from Dall-e 3 ($\approx \times 15$ params) (d) Samples from InCoDe (trained on top of Stable Diffusion v1-4). Red superscripts in attributes indicate the models that fail to capture them in the presented samples.

Inspired by the above scenario, our goal is to develop generative models that can be controlled through interaction with high-level concepts. Arguably, plain textual scene descriptions provide an intuitive interface for controlling image generation.

However, controlling image generation through text is extremely difficult, because the semantic content of an image can be represented through diverse text captions, each providing a distinct perspective. Take, once again, a photograph of a bedroom as an example. The scene could be described in terms of the objects it contains, the furniture style, the discernible colors, or the relationships among objects, since there is no canonical semantic representation.

Moreover, despite recent advances in text-conditioned image generation (Betker et al., 2023; Podell et al., 2023), constraining the generated image to a rich textual description that simultaneously encompasses numerous semantic concepts remains a challenge, as illustrated in Fig. 2(a,b,c). Although this issue has been partially addressed by subsequent work (Feng et al., 2023), image generation still struggles with composition in the presence of negations and large conjunctions (Tbl. 2).

To address these difficulties, we propose to represent the semantic content of images by sets of predefined, user-oriented and task-specific questions and their answers. This allows for interpretable representations[1] that can be easily modified by the user, while specializing a generative model to the sub-distribution of their interest based on the task (e.g. bedroom furniture distribution). Although this approach may limit the generality of the model, it prioritizes combining multiple concepts accurately.

Given an image and a set of questions to choose from, we propose to build interpretable representations by greedily selecting queries in order of information gain using an algorithm called Information Pursuit (Geman & Jedynak, 1996; Jahangiri et al., 2017; Chattopadhyay et al., 2022). Specifically, every subsequent question is selected such that its answer has maximum mutual information (among the remaining questions in the set) with the image content conditioned on the history of question-answer pairs observed so far. Consequently, we specialize our generative model to synthesize images conditioned on these question-answer based representations.

Our framework allows the user to efficiently and intuitively interact with the system and control generations by changing the answer to selected queries from the generated image's representation. Moreover, organizing the queries in order of information gain allows for a natural information-theoretic definition of "semantic similarity", where two images are said to be *k-similar* if they agree on the first $k$ most-informative question-answers.

**Paper contributions.** In this work, **(i)** We propose `InCoDe`, an information theoretic framework based on Information Pursuit (IP) for representing data in terms of concepts and using this representation to control image generation; **(ii)** We introduce a novel adaptation method for conditioning text-to-image pre-trained diffusion models on query-answer sets. **(iii)** We collected two new datasets along with sets of binary queries and answers about their content. **(iv)** We quantitatively and qualitatively validate the effectiveness of `InCoDe`, showcasing its superiority over the selected baselines

---

[1]Since they are based on human intelligible questions and their answers.

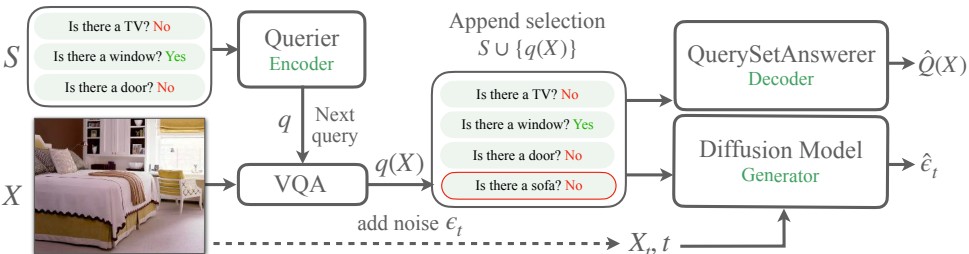

Figure 3: **Overview: pipeline of `InCoDe`**. It consists of three main modules: an *Encoder* that maps a history of query-answer pairs $S = \{(q_i, q_i(X))\}_{i=1}^{k}$ to the next asked query $q$, which is answered by providing $q$ and the reference image $X$ to an off-the-shelf VQA module. A *Decoder* that maps query-based representations to the complete query-answer set, and a diffusion-based *Generator* that predicts the added noise $\epsilon_t$ to the corrupted image $X_t$, after $t$ forward diffusion steps and conditioned on the representations.

across a diverse set of scenarios, and provide additional results in the Appendix for query-types from a different domain.

## 2 METHODS

In this section, we first formalize our choice for semantic data representation and introduce `InCoDe`, a new image generation framework, which can be controlled via compressed interpretable representations. Then, we describe architecture designs to implement the proposed framework and discuss training strategies. Finally, we explain how to use `InCoDe` to generate and edit images.

### 2.1 INCODE FRAMEWORK

We propose `InCoDe`, which uses interpretable compressed representations to provide control over image generation and semantic editing. An overview of `InCoDe`'s training pipeline is depicted in Figure 3. It consists of three main modules: an *Encoder*, which produces query-based representations by selecting elements from a given query set and a reference image to answer the selected queries; a *Decoder*, which maps these representations back to the complete query set with the corresponding answers for the reference image; and a *Generator*, which uses reverse diffusion (i.e., gradual denoising) to turn the representations into images. Next, we formally define the concept of a query set and outline the information-theoretic procedure used to organize its elements.

**Notation.** Let $\mathcal{X}$ be a set of images, $Q$ be a set of user-defined task-specific queries, and $\mathcal{A}$ be the set of all possible answers. A query $q \in Q$ is a function mapping a point in $X \in \mathcal{X}$ to a point in $a \in \mathcal{A}$, i.e. $q: \mathcal{X} \to \mathcal{A}$. In the sequel, using a slight abuse of notation, we will denote a query-answer pair $(q, a)$ for $X$ simply as $q(X)$. Finally, we denote by $Q(X)$ the set of all query-answer pairs for $X$, $Q(X) = \{q(X) \mid q \in Q\}$.

**Semantic Representation.** We describe an image $X$ with a sequence of query-answers, $D(X) = q_{1:L}(X) = [q_1(X), \ldots, q_L(X)]$, $q_i(X) \in Q(X)$, $L = |D| \leq |Q|$, where the queries are sorted in decreasing order of the information gain that they provide about all other query answers in $Q(X)$.

Given an observation $X = x^{\text{obs}}$, $D(x^{\text{obs}})$ can be generated using the Information Pursuit (IP) algorithm (see Sec. A.2 for details), modified so that at each step it selects the most informative query towards recovering the complete set $Q(x^{\text{obs}})$ from the answers in $D(x^{\text{obs}})$:

$$
\begin{aligned}
q_1 &= \text{IP}(\emptyset) = \arg\max_{q \in Q} I(q(X); Q(X)); \\
q_{k+1} &= \text{IP}(q_{1:k}(x^{\text{obs}})) = \arg\max_{q \in Q} I(q(X); Q(X) \mid q_{1:k}(x^{\text{obs}})).
\end{aligned}
\tag{1}
$$

where $I(\cdot; \cdot)$ is mutual information and $q_{1:k}(x^{\text{obs}})$ denotes the history of query-answers obtained so far. The algorithm terminates once $L$ queries have been asked, where $L$ is a hyperparameter that controls the compactness of the representation.

Following Chattopadhyay et al. (2023), we use Variational IP (VIP) to efficiently implement IP. The main idea is to learn a function, called *Querier*, directly from data such that given any set of

query-answer pairs it outputs the most informative next query, without the need to explicitly compute mutual information (which is challenging in high-dimensions). To do this, VIP employs a *Querier g*, working as an *Encoder*, and a *QuerySet Answerer*, working as a *Decoder*, as formally described next.

**Encoder.** Let $\mathbb{P}(Q(X))$ denote the power set[2] of $Q(X)$. A *Querier* function $g : \mathbb{P}(Q(X)) \rightarrow Q$ greedily selects elements from $Q$ that yield the most succinct representation of $X$ for predicting $Q(X)$. More specifically, given an observation $X = x^{\mathrm{obs}}$, the *Querier* takes as input a history of arbitrary length $k$ of queries and answers, $S = q_{1:k}(x^{\mathrm{obs}}) = \{q_1(x^{\mathrm{obs}}), \dots, q_k(x^{\mathrm{obs}})\} \subseteq Q(x^{\mathrm{obs}})$, and outputs the query $q_{k+1} \in Q$ that is the most informative about the complete set $Q(X)$ (equation 1). To do this, we will seek a query $q^*$ whose answer minimizes the KL divergence between the posterior $p(Q(X) \mid X)$ and the conditional query set distribution $p(Q(X) \mid q^*(X), S)$, where conditioning on $S$ should be read as conditioning on the event $\{x' \in \mathcal{X} \mid q_{1:k}(x') = q_{1:k}(x^{\mathrm{obs}})\}$.

**Decoder.** Since the above posterior depends on the data distribution and it is unknown, we will use a *QuerySet Answerer* $f : \mathbb{P}(Q(X)) \rightarrow \mathcal{P}_Q(\mathcal{X})$, which maps a set of query-answers to a distribution over $Q(\mathcal{X})$ and learns the posterior together with the *Querier g*. For this purpose, we adapt the VIP objective from Chattopadhyay et al. (2023) to recover the set $Q(X)$. More specifically, let $D_{\mathrm{KL}}$ be the Kullback–Leibler divergence between two probability functions, and let $S$ be a history of randomly chosen query-answer pairs. We find the querier $g$ and the queryset answerer $f$ by solving the problem:

$$\min_{f,g} \quad \mathbb{E}_{X,S} D_{\mathrm{KL}}[p(Q(X) \mid X) \parallel \hat{p}(Q(X) \mid q(X), S)]$$
$$\text{where} \quad q := g(S) \in Q, \qquad \hat{p}(Q(X) \mid q(X), S) := f(q(X) \cup S). \tag{2}$$

In order, given the observed data point $X = x^{\mathrm{obs}}$ and an observed history $S = q_{1:k}(x^{\mathrm{obs}})$, the *Querier g* selects a query $q \in Q$, evaluates it on $x^{\mathrm{obs}}$ (Visual Question-Answering [VQA] module in Figure3), and feeds the extended set $q_{1:k+1}(x^{\mathrm{obs}}) = q(x^{\mathrm{obs}}) \cup q_{1:k}(x^{\mathrm{obs}})$ to the *QuerySet Answerer*, which outputs the distribution of answers to *all* the queries in $Q$ given the partially observed set $q_{1:k+1}(x^{\mathrm{obs}})$.

Chattopadhyay et al. (2023) prove that the optimal $(f, g)$ obtained by VIP selects queries that maximize mutual information with respect to a target variable. In our appendix, we reformulate that proof with $Q(X)$ as our variable of interest. In practice, since optimization over all possible functions $f$ and $g$ is challenging, we parametrize them as neural networks: $f_\theta$ and $g_\phi$, respectively.

**Generator.** The role of the third module in `InCoDe` is to generate image $\hat{X}$, conditioned on a representation $D$ using a conditional generative model or *Generator*. Given a representation $D$, the *Generator* produces samples $\hat{X} \sim p(X \mid D)$. If the *Querier* was used to encode $D$ from a data point $X$, then, an optimal system would generate $\hat{X}$ such that the agreement between $Q(X)$ and $Q(\hat{X})$ is maximized. Notably, at inference-time, $D$ can also be hand-crafted by the user by providing their desired answers to the queries sequentially asked by the *Querier*, as illustrated in Figs. 1c and 2d.

Given the recent success in conditional generation, we propose to leverage Diffusion models for our generative module. In particular, we utilize the Denoising Diffusion Probabilistic Method (DDPM) (Ho et al., 2020), conditioned on sets of query-answers $S$. Next, we describe the optimization procedure for training the *Generator*.

In generative modeling, we seek to maximize the marginal likelihood of the data $\mathbb{E}_{x_0 \sim p(X_0)}[\hat{p}_\theta(X_0)]$ under our parametric model. This objective is often intractable. Instead, it is common to define and optimize the variational lower bound (VLB) of that quantity. In DDPM, a simplified noise-matching objective is used, which is derived from the VLB (see Sec. A.2) and simplified as:

$$\min_\theta \mathbb{E}_{X_0,t}[D_{\mathrm{KL}}(p(X_{t-1} \mid X_t, X_0) \parallel \hat{p}_\theta(X_{t-1} \mid X_t)] \approx \min_\theta \mathbb{E}_{X_0,t}\left[\|\epsilon_t - \epsilon_\theta(X_t, t)\|^2\right] \tag{3}$$

with $t \sim \mathrm{Uniform}(1, T)$, $X_0 = X$ is the clean input data, and $X_t$ is the noisy data after $t$ forward diffusion steps. Here, $\hat{p}_\theta(\cdot)$ is the DDPM's posterior, $\epsilon_\theta$ the parameterized noise estimator and $\epsilon_t$ is the Gaussian noise added to $X_0$ at time $t$ during the forward diffusion process. Please see the Appendix for the intermediate steps. As in Sec. A.2, we refer to the noisy version of $X_0$ along the diffusion process as $X_t$, with $t$ being the diffusion time and therefore the noise scale.

Here, we wish to sample from a distribution conditioned on elements of $\mathbb{P}(Q(X))$, such as histories $S$ (the sets of query-answer pairs observed during IP) or representations $D$. We incorporate these

---

[2]A power set of a set $Q$ is the set of all subsets of $Q$, including the empty set and $Q$ itself.

conditions into our objective by using the conditional distribution while preserving the objective function (Ho et al., 2020):

$$\min_\theta \mathbb{E}_{t,X,S} D_{\mathrm{KL}}[p(X_{t-1} | X_t, X_0) \,||\, \hat{p}_\theta(X_{t-1} | X_t, S)] \approx \min_\theta \mathbb{E}_{t,X,S}\big[\|\epsilon_t - \epsilon_\theta(X_t, t, S)\|^2\big] \quad (4)$$

Using query-answer pairs as conditioning variables has not been explored before. Even a modestly-sized query set $Q$ can lead to a combinatorial explosion in the possible values $S$ can take. This makes optimizing equation 4 challenging. Next, in Section 2.2 we provide architectural and training details to address this challenge.

## 2.2 ARCHITECTURE DESIGN, TRAINING, AND SAMPLING

As noted above, the three main modules of `InCoDe`, the *Querier*, *QuerySet Answerer* and *Generator*, are parameterized as neural-networks. In this section, we discuss our design choices for the scenario where $Q$ consists of binary queries about the presence/absence of semantic attributes in image $X$.

**Training the Querier and Query-Answerer.** We parameterize both the *Querier* and the *QuerySet Answerer* as fully-connected networks, which are trained jointly using the objective function in equation 2. Note that this objective requires computing an expectation over all possible histories of query-answer pairs.

Since optimizing for all query-answer pair combinations is intractable, the networks must extrapolate from limited examples. Following Chattopadhyay et al. (2023), we employ different query sampling strategies during training. At the initial stages of training, we use a **Random Strategy** where we randomly sample a history of length $M$, with $M \sim \mathrm{Uniform}(0, |Q| - 1)$. Later, as the queried selections become more reliable, we refine them by employing a **Biased Strategy**, where we sample a history of size $M$ sequentially using the *Querier*'s outputs. To answer queries at training time, we either use the dataset ground-truth (if available) or an off-the-shelf VQA model.[3]

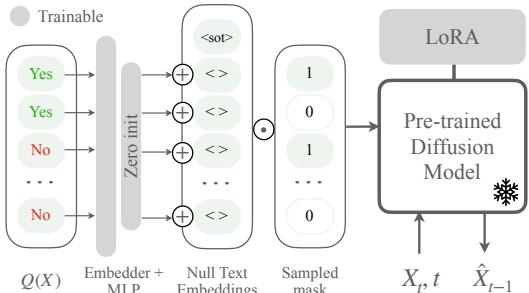

**Conditioning the Generator with Queries.** Current text-to-image models struggle to accurately generate images constrained by multiple concepts (as shown in Fig. 2). Therefore, we avoid relying on natural language for this task. In our context, a *Generator* that can effectively combine multiple concepts within a limited semantic space is preferable to one that attempts to generalize to all possible text prompts.

Figure 4: **Adapting pre-trained diffusion models to query-based conditions**. Proposed scheme to fine-tune text-to-image diffusion models with LoRA Hu et al. (2022). We introduce a small neural network (embedder plus a MLP with zero-init for the last layer) that maps queries into null-text token features. Then, we use a mask to train the model with a subset of queries.

Hence, we adapt the conditioning procedure of our diffusion-based generator to a given query set $Q$. Given a query history $S$, we encode it into a set of feature vectors. Then, we employ cross-attention layers between these feature vectors and the image features at multiple hierarchies of the diffusion model backbone (UNet), as usually done by text-to-image diffusion models (Zhang et al., 2023; Rombach et al., 2022).

Since our query-based conditioning setting is not typical in the literature, we lack access to large pre-trained diffusion models that can be readily applied to our use case. Consequently, inspired by previous work (Zhang et al., 2023; Shi et al., 2024), we propose a novel fine-tuning approach, illustrated in Fig. 4. As shown in the figure, our method uses a module with two main blocks: **(i)** A LoRA Hu et al. (2022) system with rank 32 set as a fine-tuning framework for the frozen large diffusion model, and **(ii)** A query *embedder* network, consisting of a feature embedding and an MLP with a zero-initialized final projection. During training, we embed each query-answer pair $q(X) \in Q(X)$ individually, into the dimensionality of a single text token using the *embedder* network. Then, the embedded query-answer pair is masked according to whether or not the query-answer is

---

[3]Note that a VQA model is used to obtain answers to a query given an input image, while the QuerySet Answerer predicts the answers to all queries given the answers to a subset of queries from $Q$.

present in $S$, summed to a *null-text token*[4] and fed to the UNet. Note that at the beginning of training, the UNet only sees *null-text tokens* and performs unconditional denoising. As the training advances, the network learns to adapt to the given subset of data while utilizing the conditional information in $S$, which is sampled uniformly at random. By fine-tuning an existing pre-trained text-to-image diffusion model, such as Stable Diffusion Rombach et al. (2022), our method can take advantage of the vast domain knowledge encoded in these models.

**Beyond Attribute-Based Queries.** `InCoDe` is not restricted to binary queries. Our framework supports queries gathering different types of information about a datapoint, including non-binary answers that may even be continuous or spatially-aware queries. For example, in the Appendix we provide experiments conducted with location-based queries, where a query $q_{i,j}$ requests the pixel values of an RGB image patch around the location $(i, j)$ in $X$. Location-based queries are different from attribute-based ones in two key aspects. Firstly, in this case, we can have $Q(X) = X$, that is, asking all the queries in $Q$ is the same as observing the image $X$. Secondly, since these questions are directly associated with image coordinates, they implicitly carry spatial structural information. These differences lead to a different architecture design. In this case, it is convenient to parameterize a set $S$ of location-based queries and their answers as an image, where only the patches corresponding to answered questions are visible. Thus, for location-based queries we use a *Querier* consisting of convolutional blocks. A detailed explanation and results can be found in the Appendix (Sec. A.1.1).

**Image Generation.** The procedure for image generation is straightforward. The input to the *Generator* is a representation of length $L$, either generated from a reference image (Fig. 1b), which can be modified by the user (Fig. 1c), or entirely handcrafted by the user by answering questions from the set $Q$ selected by the *Querier* (Fig. 2d). The *Querier*, starting from an empty history $S = \emptyset$, selects sequentially the most informative query $q$ from the query set $Q$, along with an answer produced by a VQA model, or the user. Next, $q(X)$ is appended to $S$ and the process is repeated $L$ times until a representation $D$ is obtained. Finally, image samples are generated by using the *Generator*, conditioned on the representation $D$ through a classifier-free sampling strategy (Ho & Salimans, 2021).

## 3 EXPERIMENTS

In this section we empirically evaluate the performance of `InCoDe` and provide analysis of its capabilities. In particular, we study (i) its effectiveness in capturing the semantic content of an image by evaluating the *Querier*'s ability to select queries that maximize information gain, as well as the faithfulness of the generated image to the provided representations; and (ii) its editing and compositional capabilities by evaluating its ability to modify or generate an image consistent with a desired set of attributes. **Please refer to the Appendix for additional results**, including location-based queries and qualitative analyses that highlight the features and limitations of `InCoDe`.

Table 1: Representations of the image size and query-set for different datasets.

| Dataset (Image size) | Query Set $Q$ | Size $|Q|$ |
|---|---|---|
| CLEVR Colors ($3 \times 64 \times 64$) | Indicator of presence of N or more objects of a particular color | 45 |
| CelebA ($3 \times 64 \times 64$) | Facial Attributes | 40 |
| LSUN Bedroom ($3 \times 512 \times 512$) | Room descriptor attributes | 58 |
| LSUN Churches ($3 \times 512 \times 512$) | Church and surroundings descriptor attributes | 44 |

**Datasets.** We selected four datasets with increasing complexity, starting from simple cases with predictable outcomes based on human intuition and progressing towards more challenging ones. Following are the datasets considered:

*CLEVR Colors*: A synthetic dataset (introduced in Johnson et al. (2016)), where objects with different attributes are placed randomly in a uniform background. For this set, we created an attribute-based query set with questions of the form: *"Are there $n$ or more objects of color $Y$?"*. This is asked for eight colors (including *any* color) and $n \in [1, 5]$ for a total of 45 queries. The only considered attribute for our purposes is the object's color, hence its name.

---

[4]a special token that represents "no text".

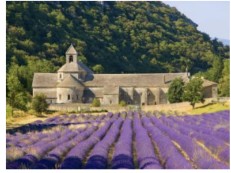 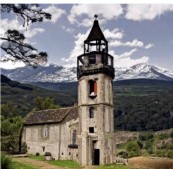 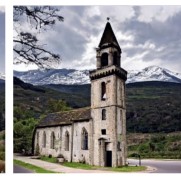 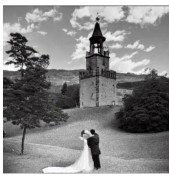 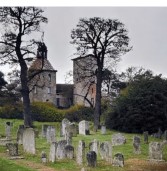

| Reference | Regeneration | Asphalt → Yes | People → Yes | Mountains → No |

Figure 5: **InCoDe offers users intuitive tools to adjust the semantic elements of generated images**. InCoDe describes an image in the query-answer space and generates a new, semantically aligned image. By changing the answer to specific queries, we can modify semantic attributes of the new images.

*CelebA*: A dataset with celebrity face images and 40 facial attributes provided with the images.

*LSUN Bedroom and LSUN Churches*: Datasets with 58 queries about the room layout and 44 queries about the elements around and belonging to each church, respectively. Both datasets were developed using off-the-shelf models, with a large language model (LLM) assisting in the creation of the query sets and a visual question answering (VQA) model (BLIP, Li et al. (2022)) generating the answers. These datasets are a key contribution of this work, filling a gap where no existing datasets meet the specific requirements of our task, and have been made publicly available.

The types and number of queries used for each of these datasets are listed in Table 1. More details can be found in the Appendix.

**Metrics.** We evaluate the effectiveness of the information acquisition strategy by predicting answers to the complete set of queries $Q(X)$, using the representations $D$ gathered by each strategy. The set $Q(X)$ is predicted with an attribute Classifier, trained to classify $Q(X)$ from random histories $S$. Accuracy and F1 score are computed with a testset with $2k$ samples in all cases, comparing predicted answers with the ground truth for each acquired query.

To evaluate compositional generation, we compute alignment to the query-answer set by running BLIP to answer queries from $Q$ on the generated images. We report accuracy and F1 score with respect to the ground-truth conditional signal, for the 10 queries with highest entropy.

**Baselines.** We compare our method with the following query-selection strategies.

- **Random sampling baseline**, where queries are sampled uniformly without replacement.
- **Decision tree with information gain by impurity criterion (DT-IC)**, where we choose the query with largest entropy (computed from the dataset), given the previous history of query-answer pairs. This baseline is the gold standard, as it seeks to maximize the entropy of the selected query given a history, which is our ultimate objective given $I(q(X); Q(X)|s) = H(q(X) \mid s) - H(q(X) \mid Q(X), s) = H(q(X) \mid s)$. However, since the entropy is estimated using empirical probabilities, DT-IC suffers from data fragmentation and long computation times for large data corpora.
- **Top K with impurity criterion (TopK-IC)**, where we select the top-k queries based on their answer's entropy in the training set.

Finally, for compositional generation experiments we compare against **(i)** Stable Diffusion V1-4 baseline text-to-image model; and **(ii)** Structured Diffusion Feng et al. (2023), a method to improve multi-concept conditioning in text-to-image models, specifically more accurate attribute binding and better image compositions. All methods are conditioned by text, and thus we provide the attribute list as concatenated representations (e.g. *"Photo of a bedroom that does not have white walls, has a window, ..."*). Note that given the restricted length of text tokens accepted by the baselines (77 tokens) this experiment is performed with only 10 attributes from the query-set, selected by descending order of entropy in the dataset. All generative models (including ours) in our experiments have SD V1-4 as the base model for fair comparison. Note that our fine-tuning method can be plugged into other text-to-image diffusion models. More details are given in the Appendix.

### 3.1 QUALITATIVE EXPERIMENTAL RESULTS

**Semantic Auto-Encoding and Concept Manipulation.** Figs. 1 and 2 illustrate qualitative results for the LSUN Bedroom dataset. In these examples, Figure 1(b) shows four images generated using the complete description with $L = 58$ query-answer pairs, while Figure 1(c) shows examples generated

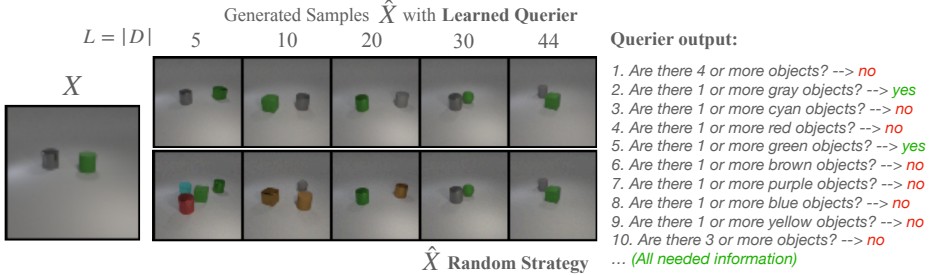

Figure 6: **Representation of `InCoDe`'s performance in CLEVR Colors dataset.** We illustrate one example of generated samples by utilizing our model's *Querier* and the random baseline. `InCoDe` generates a sequence of sensible queries and the **Generator** successfully generates an image that is semantically equivalent (as defined by our query set) to the reference image after only asking few queries.

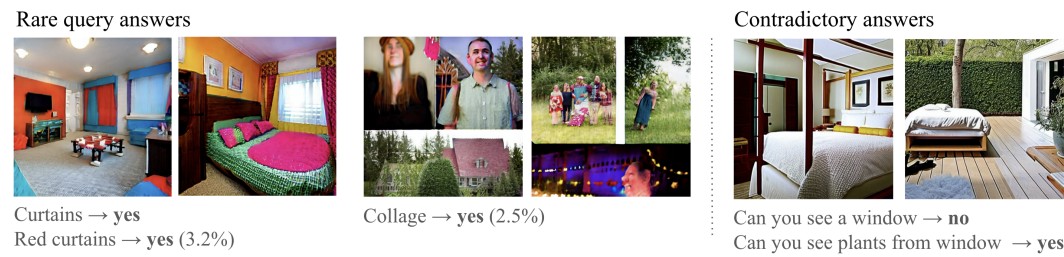

Figure 7: **`InCoDe` behaves correctly in corner cases.** Left: In case of generating concepts underrepresented in the training set, images still can capture them. Right: For apparently contradictory answers to queries, the *Generator* tries to accomplish both at the same time resulting in creative solutions.

by (left) changing one, and (right) two attributes. In Fig. 2, our method successfully generates samples based solely on a sequence of query-answers, while other methods fail.

Similarly, Fig. 5 presents an example for LSUN Churches, where `InCoDe` regenerates a reference image by first representing it in $Q(X)$, and then adjusts the answers to specific attributes.

In all cases, `InCoDe` results preserve quality while showcasing correct behavior. To appreciate similarity between generated and reference images, refer to the list of attributes for Bedroom and Churches found in Dataset details (Sec. A.3). More qualitative results can also be found in Sec. A.1.

**Querier trajectory.** In Fig. 6, we see a simple yet illustrative example of a trajectory generated by `InCoDe`, as well as one generated by a random baseline, with the corresponding generated images. The CLEVR Colors dataset is useful since it is easy for a human to see whether the output of the querier is correct or not. For instance, in the depicted case, the *Querier* begins by asking a question about the number of objects and obtains an answer 4. Next, it proceeds to ask about the colors one by one and it finds that there are 1 or more objects of gray color and green color. Therefore, there is a minimum of 2 objects. Subsequently, at iteration 10, `InCoDe` asks if there are 3 or more objects, and receives a negative response thereby completely capturing the semantics of the reference image as defined by our query set, which is only concerned with the number of objects in the image and their color. As the result, the *Generator* manages to synthesis an image that matches our reference image in semantic content (again as defined by our query set).

**Corner Cases.** Figure 7 illustrates the behavior of `InCoDe` when generating images from concepts that are underrepresented in the training set, demonstrating that these concepts are successfully captured through the proposed fine-tuning process. Additionally, the figure presents an example where `InCoDe` is faced with seemingly contradictory answers to different queries and synthesis a creative but implausible solution.

## 3.2 QUANTITATIVE EXPERIMENTAL RESULTS

**Effectiveness of the Information acquisition strategy.** The results for these experiments are illustrated in Figure 8, where we plot Accuracy and F1-score against description length, where higher values indicate better performance. Both DT-IC and TopK-IC compute the entropy and select the

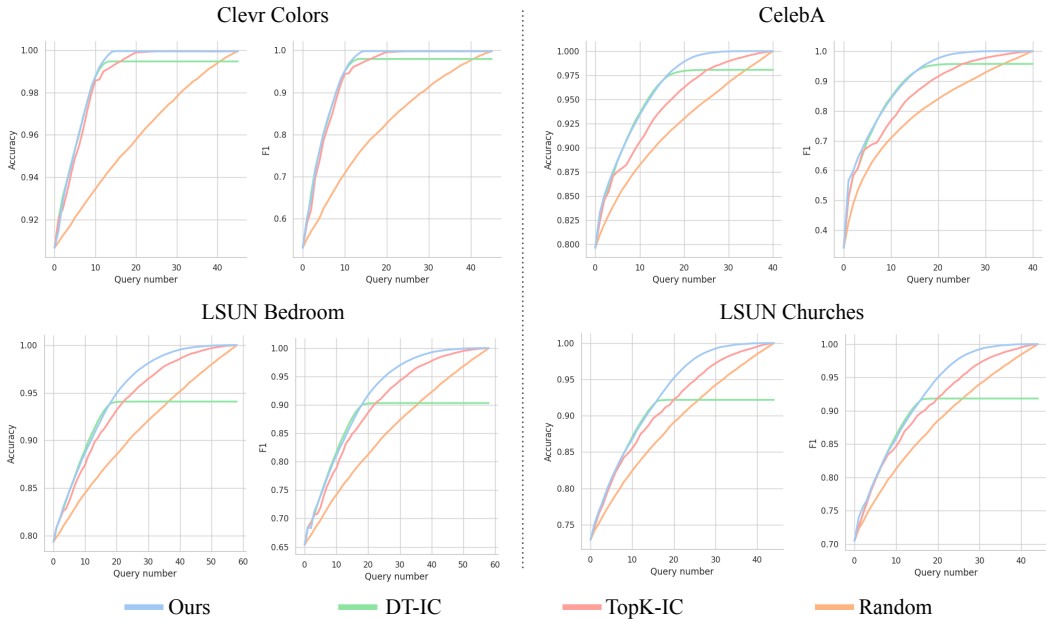

Figure 8: **Performance evaluation of `InCoDe` using attribute-based queries.** We report the Test Accuracy and F1 score of an attribute classifier on the predicted representations. `InCoDe` outperforms the baselines in all cases, with increasing gains as the datasets become more complex. DT-IC shows similar performance initially, but it suffers from data fragmentation, as well as long computation time.

query that maximizes it. The key difference is that DT-IC takes into consideration each query-answer and fragments the data accordingly. Eventually, the method runs out of data points in the training set over which computing the entropy and thus it stops selecting queries (visible by the flattening of the curve). In other words, it suffers from data fragmentation. TopK-IC does not have this issue, but it is suboptimal as it does not account for history.

This experiment shows `InCoDe` consistently selects queries that yield a better information acquisition strategy than the baselines, especially for more complex scenarios such as the LSUN datasets. In contrast, the CLEVR dataset is simple and contains a high degree of redundancy in $Q(X)$. The latter favors entropy-based methods while explaining the poor performance shown by the random strategy.

**Composable generation.** The Accuracy and F1 Score in Tbl. 2 for the LSUN datasets show the capacity of `InCoDe` to combine multiple concepts into the same generated sample. Note that the chosen metrics introduce a new source of error as BLIP can produce bad predictions.

We can observe that `InCoDe` performs significantly better than Stable Diffusion (SD V1), which is the base method upon which `InCoDe` is built. It also outperforms SD XL, which is a bigger, enhanced version of SD V1. Structured Diffusion (Feng et al. (2023)) is also based on SD; however, the modifications they propose negatively impact performance when dealing with the conjunction of numerous concepts.

Table 2: **Our method generates images that respect the attributes more often than baselines do.** Quantitative evaluation in terms of Accuracy and F1-score for the top 10 queries, according to their entropy computed from their respective trainset.

|  | LSUN Bedroom | | LSUN Churches | |
|---|---|---|---|---|
|  | Acc. | F1 | Acc. | F1 |
| SD V1 | 0.61 | 0.62 | 0.57 | 0.55 |
| SD XL | 0.58 | 0.64 | 0.59 | 0.61 |
| Stru. D | 0.52 | 0.54 | 0.51 | 0.50 |
| `InCoDe` | **0.85** | **0.84** | **0.75** | **0.72** |

## 4 RELATED WORK

**Interpretability in Machine Learning.** A large number of papers are devoted to *post-hoc* interpretability. Recent research focuses on developing more principled frameworks where interpretability is part of the model's design ((Wu et al., 2021; Bohle et al., 2021; Alvarez Melis & Jaakkola, 2018)). Another line of work learns latent semantic concepts or prototypes from data ((Sarkar et al., 2022; Nauta et al., 2021; Donnelly et al., 2022; Li et al., 2018; Yeh et al., 2020)) and produces predictions

by leveraging those concepts. Nevertheless, these learned concepts are not always interpretable to the end user. Instead, (Chattopadhyay et al., 2022) introduced IP (with subsequent work in Gadgil et al. (2024)), which produces predictions explained by interpretable query-chains, allowing the user to define intermediate representations in the form of a query-set. This guarantees by construction that the resulting query-chain explanations will be interpretable.

**Conditioning, Control and Interpretability In Diffusion Models.** Diffusion models have been used for conditional generation with great success. While early conditioning diffusion models studied class-conditional generation (Dhariwal & Nichol (2021); Ho & Salimans (2021)), recent focus has shifted to text-to-image generation, with high-quality results (Rombach et al. (2022); Saharia et al. (2022); Ramesh et al. (2022); Nichol et al. (2021)). Consequently, many works on controllability rely on the direct intervention of embedded text representations (Kawar et al. (2023); Ramesh et al. (2022); Avrahami et al. (2022); Kim et al. (2022); Feng et al. (2023)). In this case, image manipulation still relies on the ability of text-to-image models to compose multiple concepts, which often fail (Fig. 2, Tbl. 2). Alternative methods, such as those in Mokady et al. (2022); Hertz et al. (2023); Epstein et al. (2023), leverage the analysis and manipulation of cross-attention maps between text tokens and U-Net features for control. Regardless of the text-based method, when control does not directly involve text prompts, interpretability and subsequent manipulation typically rely on post-hoc analysis of the network's features.

Beyond text-based conditioning, several efforts have focused on using alternative control signals to guide the generative process. Works such as Zhang et al. (2023); Zheng et al. (2023); Du et al. (2023); Meng et al. (2022); Li et al. (2023); Yang et al. (2023); Huang et al. (2023) are representative examples of approaches for controlling image generation by means of spatially grounded inputs. These signals include contours, bounding boxes, masked images, depth maps, and sketches, often integrated with textual inputs. Interpretability relies on the user's ability to understand these control signals, which is not guaranteed. Control of the generated outputs has also been engineered directly on U-net features Kwon et al. (2023), or on abstract latent codes Preechakul et al. (2022), both requiring post hoc interpretation.

Our work presents a principled framework reminiscent of model inversion Ramesh et al. (2022); Gal et al. (2023); Mokady et al. (2022); Kwon et al. (2023) or semantic compression Preechakul et al. (2022); Koh et al. (2020); Kodirov et al. (2017). However, while these works find representations of the semantic content of a data point, they are often not interpretable by design, and when they are Koh et al. (2020); Kodirov et al. (2017), they are not compressed (they have no measure of relevance) or principled. We propose a general method that greedily selects elements from an interpretable-by-design query-answer set that are most informative, and generates new data conditioned on these representations while allowing for targeted semantic modifications.

## 5 CONCLUSION

This work introduces Interpretable Compressed Descriptions for Image Generation (`InCoDe`), a novel framework that leverages Information Pursuit (IP) to effectively represent data and guide image generation based on user preferences. `InCoDe` generates images consistent with a succinct and meaningful representation $D$, given by a sequence of user-defined questions and answers, which are chosen sequentially to maximize mutual information between $D$ and the complete Query-Answer set $Q(X)$. In this way, `InCoDe` prioritizes the most relevant queries and provides an intuitive, interactive interface for generating customized images, while addressing challenges of current generative models by building upon them. Through validation across multiple scenarios, `InCoDe` demonstrates superior performance over existing methods, offering an efficient solution for domain-specific applications such as bedroom decoration. This work paves the way for more personalized and interpretable-by-design generative models, enhancing both user experience and practical utility.

The proposed framework requires having a query set. This can be perceived as both a strength and a limitation. It is a strength, since the designer is free to select the set of semantic concepts that are important for the intended generation task. Yet, building this set can be tedious. Modern large language models can alleviate this task. We have used this approach to create the queries for the LSUN Bedroom and Churches datasets. However, human supervision and prompting is still necessary. This raises the research question of designing a framework that allows users to define the semantic domain of a query set without the need for direct supervision of the query set itself.

## 6 ETHICS STATEMENT

The development of generation methods such as `InCoDe` offers significant benefits, such as enhanced personalization and domain-specific applications, but also presents risks. Without proper safeguards, generative models can be misused in harmful ways, such as spreading disinformation or creating offensive content. As they become more accessible, it is important to balance their innovative potential with ethical considerations, ensuring that the technology is used responsibly and transparently.

In particular, one major issue is the potential for bias. If the query set used by `InCoDe` is not diverse or representative, it may result in the reinforcement of stereotypes or exclusion of certain groups. Thus, adequate oversight ensuring that the query set is ethically designed is key to mitigating this risk and avoid harmful or misleading content.

## 7 ACKNOWLEDGMENTS

This work was supported in part by NSF grant CNS-2038493, ONR grant N00014-21-1-2431, ONR MURI contract 503405-78051, ARO MURI contract W911NF-17-1-0304, and the NSF–Simons Research Collaboration on the Mathematical and Scientific Foundations of Deep Learning (NSF grant 2031985, Simons grant 814201).

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

# A APPENDIX

In this appendix, we will provide additional information to complement the main paper. We structure it as follows: After a brief summary of the notation, in Section A.1 we provide and analyse additional results for all datasets. In particular, in Section A.1.1, we discuss results on a different type of queries that are location-centered. In A.1.2, we present results that compliment those in the main body of the paper. Next, in SectionA.2 we summarize background material on Information Pursuit and Diffusion Denoising Models. In Section A.3 we provide additional details for each of our experiments; Section A.4 is used to proof Proposition 2.1 of the location-based experiments described in Sec. A.1 of this appendix and a justification of the training objective. Finally, in Section A.5, we discuss the broader and ethical impact of our work.

**Notation**   We summarize the notation for the main concepts used along this appendix.

- $p(X_0)$: Real data distribution
- $\hat{p}(X_0)$: Our model of the data distribution.
- $X_0$: Data-point sampled from a real data distribution.
- $t$: Time step in our forward diffusion process. It represents the noise level. I.e. we add small amount of Gaussian noise to the sample in T steps. This produces a set of noisy samples $(X_1, \ldots X_T)$
- $X_t$: Noisy sample at noise level t.
- $p(X_t \mid X_{t-1})$: Forward diffusion process distribution.
- $p(X_{t-1} \mid X_t, X_0, s)$: Reverse diffusion process true posterior distribution.
- $\hat{p}(X_{t-1} \mid X_t, s)$: Our model of the reverse diffusion process posterior distribution. $\hat{p}$ is also used as the denoising mapping function. The subscript $\theta$ is used to indicate the optimizable parameters of the model.
- $\beta_t$: Variance schedule.
- $c$: Arbitrary condition
- $\{q, q(X_0)\}$: Query and Queryset Answer respectively.
- $S$: Query-Answer history random variable. With $s$ as a Query-Answer history realization.
- $\epsilon_t$: Gaussian noise term added to the clean sample $X_0$ in order to obtain $X_t$.
- $\epsilon_\theta(X_t, t, c)$: Estimated noise at step $t$. Output of one forward pass in our diffusion model (with parameters $\theta$). The network is conditioned to the current noisy image $X_t$, the current noise level $t$, and potentially another condition $c$, that should steer our generative model towards $p(X_0|c)$.

## A.1 MORE RESULTS

In this section we analyse additional results from the remaining datasets, including those with location-based query sets.

### A.1.1 LOCATION-BASED EXPERIMENTS

As noted in the main body, `InCoDe` is not restricted to textual queries. A query can gather different types of information about a datapoint, such as location-based queries, where a query $q_{i,j}$ requests an RGB image patch around the location $(i, j)$ in $X$. In this case, we can have $Q(X) = X$ by allowing to ask enough queries to retrieve every image pixel. Since these questions carry spatial structural information, we can parameterize a set $S$ as an image where only the patches corresponding to answered questions are visible and we use a *Querier* consisting of a set of convolutional blocks.

In the case of $Q(X) = X$, where the answers to queries are portions of the image itself, the denoising objective of DDPM can be used to optimize our *Querier*, avoiding the need for a *QuerySet Answerer*, jointly training the *Querier* and *Generator* in an end-to-end fashion.

Equation (4) can be rewritten as:

$$\min_{g,f} \quad \mathbb{E}_{t,X_0,S} D_{\text{KL}}[p(X_{t-1} \mid X_t, X_0) \mid\mid \hat{p}_\theta(X_{t-1} \mid X_t, q(X_0), S)]$$
$$\text{where} \quad q = g(S) \in Q, \qquad \hat{p}_\theta(X_{t-1} \mid X_t, q(X_0), S) = f(q(X) \cup S) \tag{A1}$$

We formalize the above procedure as follows:

**Proposition A.1.** *Let $(f^*, g^*)$ be an optimal solution to (A1). We define the optimization problem as:*

$$\max_{\hat{p} \in \mathcal{P}_\mathcal{X}, q \in Q} I(q(X_0); X_1 \mid s) - \mathbb{E}_{t,X_0,s} D_{\text{KL}}[p(X_{t-1} \mid X_t, q(X_0), s) \mid\mid \hat{p}(X_{t-1} \mid X_t, q(X_0), s)]. \tag{A2}$$

*Then, there exists an optimal solution $(\hat{p}_s^*, q_s^*)$ to the above objective for any realization $S = s$ such that $p(S = s) > 0$ such that $q_s^* = g^*(s)$ and $\hat{p}_s^* = f^*(X_t, \{q_s^*, q_s^*(X_0)\} \cup s)$.*

The proof of proposition A.1 can be found in A.4, and it draws inspiration from Chattopadhyay et al. (2022). Note that for an optimal solution $(\hat{p}_s^*, q_s^*)$, the KL divergence term would be 0, and the *Querier* would choose the query that maximizes the mutual information term between the first latent variable $X_1$ of the diffusion process and any given subset $S = s$ of $Q(X)$. While the objective is to maximize $I(q(X); X \mid s)$, this quantity is undefined for a continuous $X$. A common strategy to make mutual information well defined is to add a small Gaussian noise to $X_0$ Saxe et al. (2018). Hence, we seek to maximize $I(q(X_0); X_1 \mid s)$ instead. In section 2.2 we provide a practical methodology to solving the presented optimization problems, by means of neural-network architectures.

**Datasets.** We selected datasets with increasing complexity to assess a range of scenarios, starting from simple cases with predictable outcomes based on human intuition and progressing towards more challenging scenarios. We include results for MNIST (LeCun et al., 1998): A database of images of hand-written digits; CLEVR Johnson et al. (2016): A synthetic dataset, where objects with different attributes are placed randomly in a uniform background. For CLEVR, we create an attribute set that answers the following query-set format: *"Are there $n$ or more objects of color $Y$?"*. This is asked for eight colors (including *any* color) and $n \in [1, 5]$ for a total of 45 queries; CelebA: A dataset with celebrity face images and 40 facial attributes; and LSUN Bedroom Attributes, a large dataset with bedroom images and their descriptions as a set of 58 binary attributes consisting on answers provided by BLIP Li et al. (2022) to human-crafted queries. The latter is a contribution of this work, given the lack of existing datasets that meet our task's requirements, and it will be released for public use. More details can be found in the Appendix. The types of queries used for each of these datasets are listed in Table 3.

**Metrics.** We assess performance using different metrics, depending on the type of queries utilized. For *location-based* queries, aiming to impute missing values in an image, we employ MSE and LPIPS Zhang et al. (2018) (measuring perceptual similarity through VGG network features), computed between the ground-truth and 200 generated samples, one per test example, for description lengths

Table 3: Descriptions of the image size and location-based query-set for different datasets.

| Dataset (Image size) | Query Set $Q$ | Size $|Q|$ |
|---|---|---|
| MNIST (LeCun et al., 1998) ($1 \times 32 \times 32$), | pixel intensities in $3 \times 3$ stride 1 | 900 |
| CelebA Face ($3 \times 64 \times 64$) | pixel intensities in $8 \times 8$ patches with stride 4 | 225 |
| Clevr Johnson et al. (2016) ($3 \times 64 \times 64$), | pixel intensities in $8 \times 8$ patches with stride 4 | 225 |

of $|D| = \{0, 1, 2, 5, 10, 20, 40, 60\}$ (for CLEVR and CelebA) and $|D| = \{0, 2, 5, 10, 20\}$ (MNIST). The *Generator* used for evaluation was trained on random histories $S$ for fairness across baselines.

For experiments using *attribute-based* queries, we predict answers to the full set of queries $Q(X)$ with a *QuerySet Answerer*, trained to classify $Q(X)$ from random histories $S$. Accuracy and F1 score are computed with a testset with $2k$ samples in all cases, comparing predicted answers with the ground truth.

**Baselines** For location-based query sets, where answers are not binary and impurity computation is intractable, we employ two hand-crafted patterns that emulate human intuition, as introduced in Rangrej et al. (2022): **(iv)** Spiral, where patches are selected in spiral starting from the center of the image, and **(v)** Cross, with patches selected with a cross pattern. Baselines are described in higher detail in the Appendix.

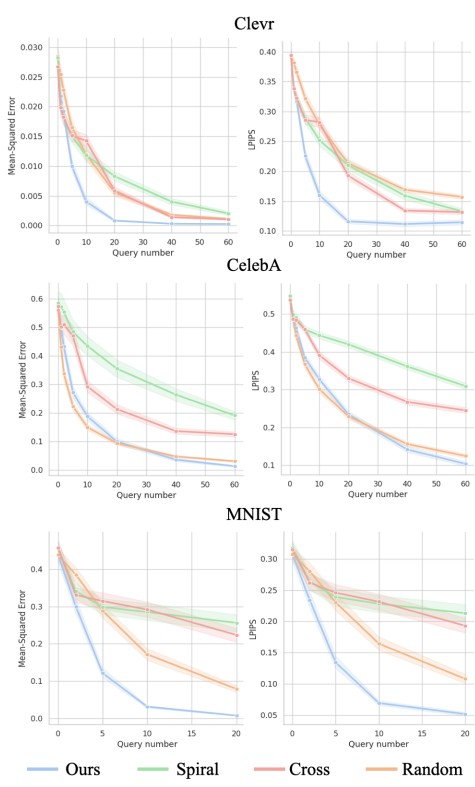

Figure 9: **Performance evaluation of `InCoDe` using location-based queries.** We measure the reconstruction error and perceptual similarity (LPIPS). `InCoDe` outperforms the baselines in both datasets.

**Quantiative results:** In Fig. 9, For these experiments we report reconstruction metrics in Figure 9, where smaller values indicate better performance. The horizontal axes indicate the number of asked queries. `InCoDe` clearly outperforms the baselines in both MNIST and CLEVR datasets, where there is abundant redundancy to be exploited. For instance, both datasets have uniform backgrounds. As the main content is located in the center of the image, spiral and cross baselines have relatively good performance in the initial steps. However, exploration is needed to disambiguate the remaining appearance and thus the random baseline gets better as more queries are asked. In the case of CelebA, we observe that the random baseline slightly surpasses `InCoDe` for fewer than 20 queries. We attribute this behavior to two main factors. First, this baseline selects queries with the same random strategy that was used to pre-train the *Generator*. Second, CelebA samples contain useful information in most of the image, and thus a highly exploratory method can be especially effective.

**Qualitative results:** We can see in the side (a) of Fig. 10 a sequence of patch queries selected by our *Querier* for a sample of Clevr. The images in this dataset consist of a uniform background with a set of objects of different sizes located around the center. We can see how the *Querier* initially selects a patch in the center of the image and then explores around it as more objects are glimpsed. It is interesting to see that for $L = 5$, the *Querier* has seen the 3 objects. However, the generator creates a 4th object, which is does not exist. This example illustrates the potential of the *Generator* to create diverse samples, while respecting the semantic content of description $D$, and the overall data distribution. In the right side

(b) of the figure, we the same illustration for a CelebA example. Here, the *Querier* selects patches in the main features of the face, the clothes and the contour of the head, which defines the shape and provides information about both the subject and the background.

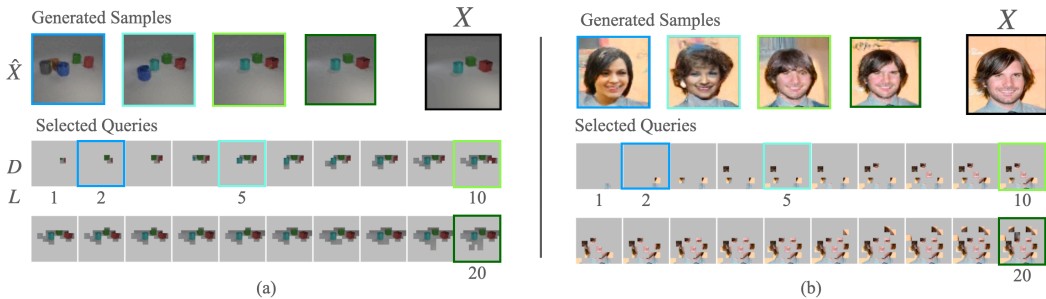

(a)                                        (b)

Figure 10: **InCoDe selects meaningful patches according to the data distribution** Illustrated in the left side (a) we see a sequence of patch queries selected by our *Querier* with their associated generated samples (same color border). The *Querier* initially selects a patch in the center of the image and then explores around it as more objects are glimpsed. In the right side (b) of the figure, we show an example for a CelebA. Here, the *Querier* selects patches in the main features of the face, the clothes and the contour of the head.

Fig.11 illustrates `InCoDe`'s performance when using *location-based* queries. The querier selects patches that contain most information about the digit. The identity of the digit is quickly captured after $|D| = 5$ patches, and further steps refine the appearance to match the reference image.

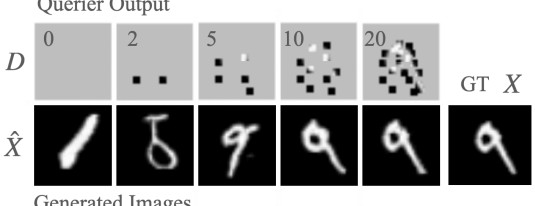

Figure 11: **InCoDe selects the most informative image patches towards reconstructing a reference.** Samples generated using descriptions with *location-based* queries of length 0, 2, 5, 10 and 20, with the MNIST dataset. The Querier selects highly informative patches according to the previous history, quickly predicts the digit's identity, and refines its appearance.

### A.1.2   ATTRIBUTE-BASED EXPERIMENTS

The following results illustrate qualitatively the behavior of our model, including corner cases.

**LSUN Bedroom more generated samples:**   In Fig. 17, we display two examples of a given set of attributes (query-answer pairs) and their corresponding image examples in the dataset. For visualization purposes we choose combinations of attributes that have exactly 4 corresponding examples in the dataset. Note that in most cases samples have a unique combination of attributes. We observe how the generated samples by our dataset are semantically similar, while appearance at the pixel level varies widely.

Fig. 12 showcases how generated samples of LSUN Bedroom dataset using *attribute-based* queries align to the reference as the number of selected queries increases. After 30 queries, the generated sample resembles image the reference just as much as when asking all queries in the query set.

**LSUN Bedrom more editing results:**   We provide additional editing results in Fig. 16. We can see how modifying one attribute does not change the overall structure of the generated sample, while it does add or subtract that particular attribute from the image.

**Trajectories:**   In Fig. 13 we present a trajectory generated by our *Querier*, along with the corresponding `InCoDe` outputs at description lengths of 5, 10, and 15. The results demonstrate alignment with the majority of answers while producing images of high quality.

**Contradictory Answers:**   in Fig. 15, we illustrate one particular case. How does the model interpret the condition in case two queries have contradictory answers. This should rarely happen when answers represent a real datapoint. However, a user could input contradictory answers, or the VQA model could have a defective output. If contradiction occurs, one or the other attribute is disregarded. This is made obvious in the figure, for queries "Is a window present in the image? **No**" and "Are plants visible from the window? **Yes**".

Reference        5        10

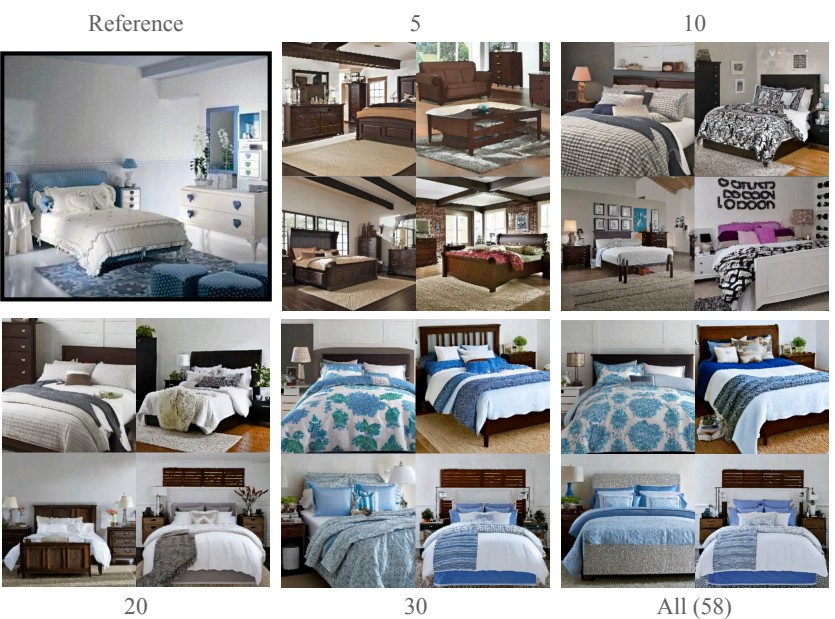

20        30        All (58)

Figure 12: **InCoDe generates samples according to a semantic description given at different levels of detail.** This figure illustrates how the descriptions chosen by our model generate samples that increasingly mirror the reference (top-left).

|D|=5        |D|=10        |D|=15        Querier output

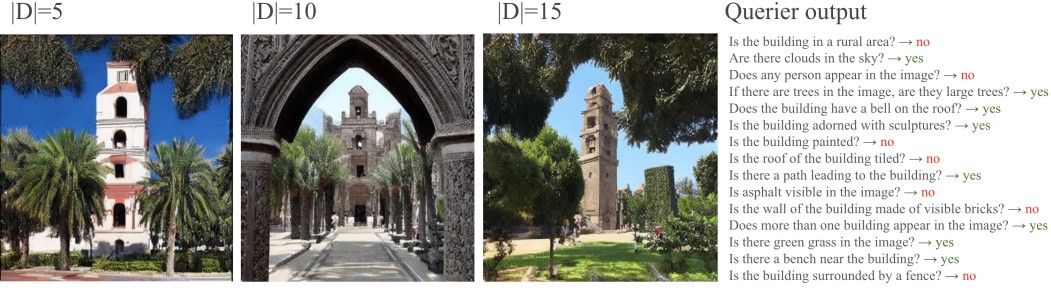

Figure 13: **InCoDe Example of a trajectory created by our Querier.** The corresponding images generated with different description lengths respect most attributes.

**Underrepresented attributes:** Fig. 14 illustrates the behavior of InCoDe when individually asked about queries with an underrepresented answer in the dataset. We observe that our approach can still capture them. However, this experiment also brings to light the limitations of our pipeline. First, we observe that the presence of a Hunting trophy is rare and disregarded by our model. If we look deeper, we conclude that the BLIP model used to answer the queries hallucinates this particular attribute, and thus the model cannot properly capture it. We also observes that the image quality suffers when only one or two queries are visible to the generative model.

**CelebA with Facial attributes:** We illustrate in Fig. 18 the performance of InCoDe for the CelebA dataset. Similarly to the previous example, we display the list of queries selected by our trained *Querier*. In this case, it is harder to judge the correctness of the sequence intuitively. However, the first three questions likely have very high entropy in the dataset. Moreover, we see that the generated images using our strategy resemble the reference at a faster pace than when using a random selection strategy.

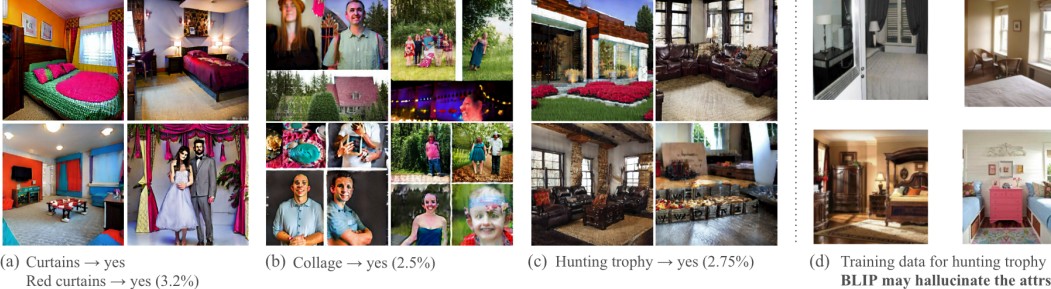

(a) Curtains → yes
    Red curtains → yes (3.2%)
        (b) Collage → yes (2.5%)
        (c) Hunting trophy → yes (2.75%)
        (d) Training data for hunting trophy
                **BLIP may hallucinate the attrs**

Figure 14: **Attributes in the dataset may be respected even when unfrequent in the dataset, however in some cases they are disregarded.** We illustrate this behavior with the following examples: (a) Generated images with active attributes indicating the presence of "red courtains are present in the room", (b) "Image is a collage" and (c) "hunting trophy is present in the room". All of them with rare presence in the training set. We observe that in some cases the attribute may be respected, with the presence of failure cases. For instance, in (c) Hunting trophys are not seen in the scene. We argue that this can also be an effect of (d), where the BLIP model used to answer the queries may sometimes hallucinate attributes and give wrong responses, as seen by retrieving examples for "hunting trophy". **We also observe that the quality of the generated images suffers slightly when the Generator is conditioned on one or two attributes only, with the rest masked out.**. Human figures also have bad quality but this is a common problem with the version of Stable-Diffusion that we are using as base model.

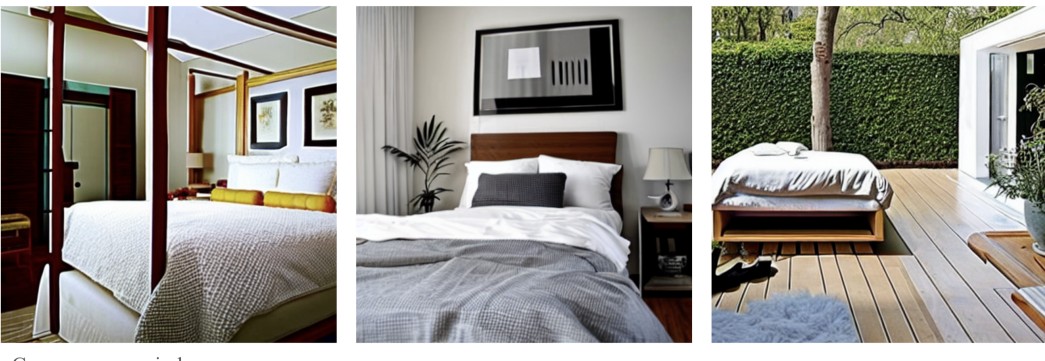

Can you see a window → **no**
Can you see plants from the window → **yes**

Figure 15: **Contradictory attributes lead to valid images, but either one attribute or the other are disregarded.** In this particular case, we see how we indicate that no window should be visible, while plants should appear through a window. The result often shows plants in an environment without windows, but obviously the conjunction of both attributes is not exactly present, as it is impossible. Note that none of the training examples (assuming VQA method being flawless) should not have such combination present.

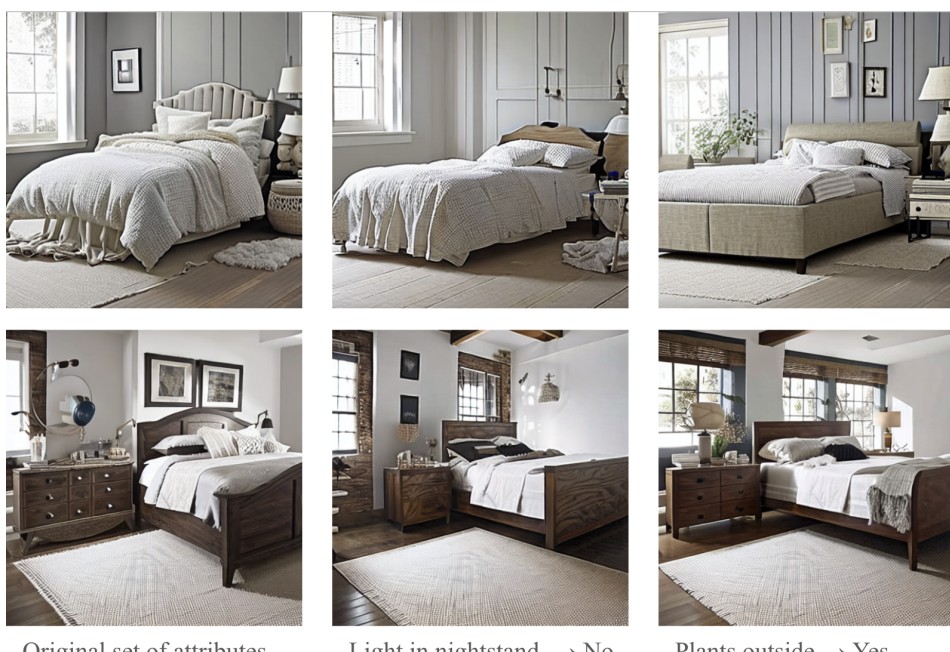

|                          |                              |                        |
|:------------------------:|:----------------------------:|:----------------------:|
| Original set of attributes | Light in nightstand → No   | Plants outside → Yes   |

Figure 16: **Examples of targeted modifications**. (left) We show two samples generated from the same set of query-answers. (middle) We edit the answer of "Is there a nightstand light in the table?" to "No". (right) We edit the answer of "Can you see plants from the window?" to "Yes".

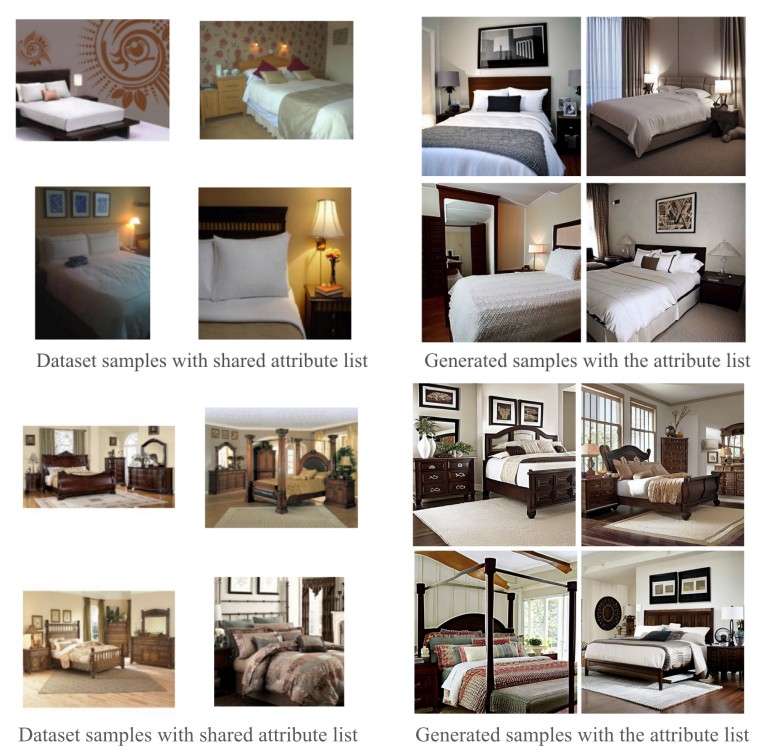

| Dataset samples with shared attribute list | Generated samples with the attribute list |
|:------------------------------------------:|:-----------------------------------------:|

| Dataset samples with shared attribute list | Generated samples with the attribute list |
|:------------------------------------------:|:-----------------------------------------:|

Figure 17: **Our method generates high-quality samples, substantially different from the training examples**. We show all examples in our training and testing set for a particular set of attribute responses. We then use these attributes to generate samples and we conclude that they are semantically similar while varying widely in appearance. We include one failure case (right figure, bottom-left).

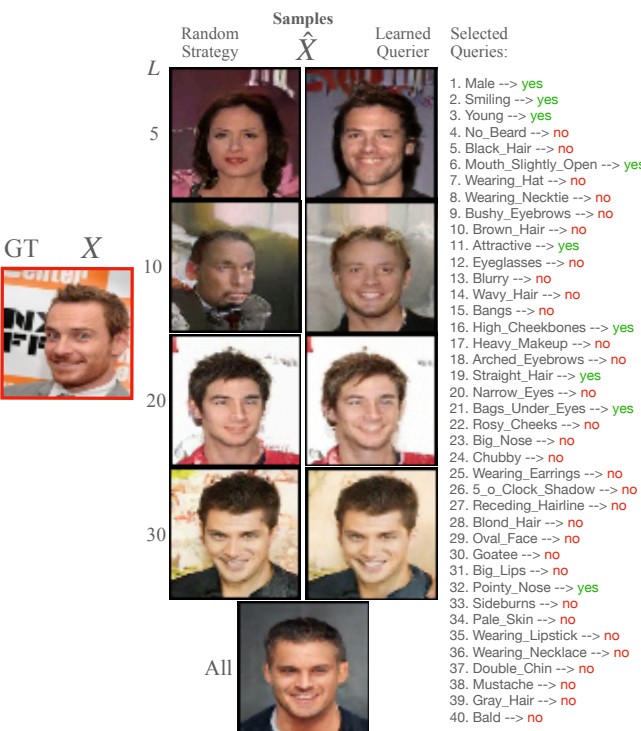

Figure 18: **InCoDe selects meaningful queries to reconstruct an image in terms of the query set**. On the right, we see a list of queries selected by our trained *Querier*. The first three questions likely have very high entropy in the dataset. The generated images (center) using our strategy converge faster towards the reference appearance (left) than the random selection strategy.

## A.2 BACKGROUND

### A.2.1 INFORMATION PURSUIT AND ITS VARIATIONAL CHARACTERIZATION

Information Pursuit (IP) was proposed as a framework for explainable prediction by choosing explainable queries Chattopadhyay et al. (2022). Its main idea is to make a prediction of a target variable $Y$ from input data $X \in \mathcal{X}$, by sequentially asking and answering queries about $X$.

In practice, only a small number of question-answer pairs might be enough to make a prediction. Thus, IP aims to construct the shortest sequence of questions that is most informative to predict $Y$, by selecting the next query to maximize the mutual information with $X$, building on the information gained from previous queries. Given a set of possible queries $Q$, and an observation $X = x^{\text{obs}}$, the IP algorithm is as follows:

$$q_1 = \text{IP}(\emptyset) = \arg\max_{q \in Q} I(q(X); Y);$$

$$q_{k+1} = \text{IP}(q_{1:k}(x^{\text{obs}}) \quad\quad\quad (A3)$$

$$= \arg\max_{q \in Q} I(q(X); Y \mid q_{1:k}(x^{\text{obs}})).$$

where $q(x)$, with a slight abuse of notation, denotes that the answer to query $q$ for the data $X = x$ is $q(x)$, and $q_{k+1} \in Q$ refers to the new query selected by IP at step $k + 1$, based on the query history (denoted as $q_{1:k}(x^{\text{obs}})$), and $I$ denotes mutual information.

IP can be carried out by learning the distribution $p(Q(X), Y)$ from the data by using generative models and MCMC sampling to estimate the mutual information terms Chattopadhyay et al. (2022). However, MCMC sampling is computationally expensive. To overcome this challenge, Chattopadhyay et al. (2023) proposed a variational characterization of IP, noting that generative models are only a means to an end. Thus, they introduced a function called *querier*, mapping the observed histories, $q_{1:k}(x^{\text{obs}})$, to the most informative next query $q_{k+1} \in Q$. They showed that this most informative query is exactly the query $q^*$ whose answer will minimize the KL divergence between the conditional label distribution $p(Y \mid X)$ and the posterior $p(Y \mid q^*(X), q_{1:k}(x^{\text{obs}}))$.

Based on this insight, an optimization problem is designed to perform IP as follows. Let $Q(x) = \{q(x) \mid q \in Q\}$ be the query-answer set containing all inquiries about the data $x$ and their corresponding answers. Let $\mathbb{K}(x) = \mathbb{P}(Q(x))$, be the power set[5] of $Q(x)$ with all possible query histories. Define $\bar{\mathbb{K}} := \cup_{x \in \mathcal{X}} \mathbb{K}(x)$; let a classifier $f : \bar{\mathbb{K}} \to \mathcal{P}_{\mathcal{Y}}$ be a function mapping arbitrary query-answer sequences to a distribution over $\mathcal{Y}$ and the querier $g : \bar{\mathbb{K}} \to Q$ be a function mapping arbitrary query-answer sequences to a query $q \in Q$. Then, the variational objective for IP is given by the optimization problem:

$$\min_{\theta, \phi} \quad \mathbb{E}_{X,S} \left[ D_{\text{KL}} \left( p(Y \mid X) \| p'_\theta(Y \mid q(X), S) \right) \right]$$

$$\text{subject to} \quad q = g_\phi(S) \in Q \quad\quad\quad (A4)$$

$$\hat{P}(Y \mid q(X), S) = f(q(X) \cup S),$$

where $D_{\text{KL}}$ indicates the Kullback–Leibler divergence between two probability functions and $S$ is a random set of query-answer pairs taking values in $\bar{\mathbb{K}}$, and conditioning on $q(X)$ should be read as conditioning on $\{x \in \mathcal{X} \mid q(x) = q(X)\}$. Given $S = s$ and $X = x^{\text{obs}}$, the querier $g_\phi(\cdot)$ chooses a query $q \in Q$, evaluates it on $x^{\text{obs}}$ and passes $q(x^{\text{obs}})$ to the classifier. Then, the classifier makes a prediction based on $s$ appended with this additional query-answer $q(x^{\text{obs}})$. This is implemented by parameterizing the querier and classifier by neural networks, with parameters $\theta$ and $\phi$, respectively, and a random set of query-answer pairs $S$. For futher details refer to Chattopadhyay et al. (2023).

### A.2.2 DIFFUSION DENOISING MODELS

Here, we provide a detailed overview of the formulation of Gaussian diffusion models from Ho et al. (2020).

There are two main stages in diffusion-based generative modeling. The first stage is the *forward diffusion* process. It gradually adds a small amount of Gaussian noise to a clean sample from the

---

[5]A power set of a set $Q$ is the set of all subsets of $Q$, including the empty set and $Q$ itself.

source distribution $X_0 \sim p(X)$ to create a sequence of noisy samples $X_0, \ldots, X_t, \ldots X_T$. The amount of added Gaussian noise is defined by a variance schedule noted by $\beta_t$:

$$p(X_t|X_{t-1}) \coloneqq \mathcal{N}(X_t; \sqrt{1-\beta_t}X_{t-1}, \beta_t \mathbf{I}) \tag{A5}$$

As seen the transition probability is parametrized as a Gaussian distribution. The Gaussian Markov proces has good properties. We can easily obtain a noisy sample at an arbitrary step $t$ of the diffusion process given a clean sample $X_0$. Instead of repeatedly applying $p$ until the desired $t$, we simply need to accumulate the noise scales given our schedule:

$$p(X_t|X_0) = \mathcal{N}(X_t; \sqrt{\bar{\alpha}_t}X_0, (1-\bar{\alpha}_t)\mathbf{I}) \tag{A6}$$

$$= \sqrt{\bar{\alpha}_t}X_0 + \epsilon\sqrt{1-\bar{\alpha}_t}, \ \epsilon \sim \mathcal{N}(0, \mathbf{I}) \tag{A7}$$

$$\text{with} \quad \alpha_t \coloneqq 1 - \beta_t \quad \text{and} \quad \bar{\alpha}_t \coloneqq \prod_{s=0}^{t} \alpha_s \tag{A8}$$

with $1-\bar{\alpha}_t$ indicating the noise variance at an arbitrary step $t$.

This is followed by the *reverse diffusion* process, which starts with a sample from isotropic Gaussian noise, $X_T$, and incrementally removes noise towards generating a true sample from the source distribution, thus reversing the forward process. We are thus interested in approximating the true posterior of the reverse diffusion process. By means of the Bayes theorem, we find that the true posterior conditioned on $X_0$ is also Gaussian with mean $\tilde{\mu}_t(X_t, X_0)$ and variance $\tilde{\beta}_t$:

$$\tilde{\mu}_t(X_t, X_0) \coloneqq \frac{\sqrt{\bar{\alpha}_{t-1}}\beta_t}{1-\bar{\alpha}_t}X_0 + \frac{\sqrt{\alpha_t}(1-\bar{\alpha}_{t-1})}{1-\bar{\alpha}_t}X_t \tag{A9}$$

$$\tilde{\beta}_t \coloneqq \frac{1-\bar{\alpha}_{t-1}}{1-\bar{\alpha}_t}\beta_t \tag{A10}$$

$$p(X_{t-1}|X_t, X_0) = \mathcal{N}(X_{t-1}; \tilde{\mu}(X_t, X_0), \tilde{\beta}_t\mathbf{I}) \tag{A11}$$

However, we wish to approximate the posterior $p(X_{t-1}|X_t)$, without access to the sample $X_0$. This term cannot be computed exactly but it is assumed to have a Gaussian form when the step-size $\beta_t$ is sufficiently small. We approximate the posterior with a model $\hat{p}_\theta(X_{t-1}|X_t)$ parametrized by a neural network with parameters $\theta$. In particular, the model estimates the mean $\mu_\theta$ and a diagonal covariance matrix $\Sigma_\theta$ of a Gaussian distribution:

$$\hat{p}_\theta(X_{t-1}|X_t) \coloneqq \mathcal{N}(X_{t-1}; \mu_\theta(X_t, t), \Sigma_\theta(X_t, t)) \tag{A12}$$

In order to learn the true distribution $p(X_0)$, we can optimize the following variational lower-bound $L_{\text{vlb}}$ for $\hat{p}_\theta(X_0)$, instead of directly approximating $p(X_0)$:

$$-\log\hat{p}_\theta(X_0) \leq -\log\hat{p}_\theta(X_0) + D_{\text{KL}}(p(X_{1:T}|X_0)\|\hat{p}_\theta(X_{1:T}|X_0)) \tag{A13}$$

$$= -\log\hat{p}_\theta(X_0) + \mathbb{E}_{X_{1:T} \sim p(X_{1:T}|X_0)}\left[\log\frac{p(X_{1:T}|X_0)}{\hat{p}_\theta(X_{0:T})/\hat{p}_\theta(X_0)}\right] \tag{A14}$$

$$= -\log\hat{p}_\theta(X_0) + \mathbb{E}_p\left[\log\frac{p(X_{1:T}|X_0)}{\hat{p}_\theta(X_{0:T})} + \log\hat{p}_\theta(X_0)\right] \tag{A15}$$

$$= \mathbb{E}_p\left[\log\frac{p(X_{1:T}|X_0)}{\hat{p}_\theta(X_{0:T})}\right] \tag{A16}$$

$$\text{Let } L_{\text{VLB}} = \mathbb{E}_{p(X_{0:T})}\left[\log\frac{p(X_{1:T}|X_0)}{\hat{p}_\theta(X_{0:T})}\right] \geq -\mathbb{E}_{p(X_0)}\log\hat{p}_\theta(X_0) \tag{A17}$$

$L_{\text{vlb}}$ can be decomposed into three main terms:

$$L_{\text{VLB}} = \mathbb{E}_{p(X_{0:T})}\Big[ \log \frac{p(X_{1:T}|X_0)}{\hat{p}_\theta(X_{0:T})} \Big] \tag{A18}$$

$$= \mathbb{E}_p\Big[ \log \frac{\prod_{t=1}^T p(X_t|X_{t-1})}{\hat{p}_\theta(X_T)\prod_{t=1}^T \hat{p}_\theta(X_{t-1}|X_t)} \Big] \tag{A19}$$

$$= \mathbb{E}_p\Big[ -\log \hat{p}_\theta(X_T) + \sum_{t=1}^T \log \frac{p(X_t|X_{t-1})}{\hat{p}_\theta(X_{t-1}|X_t)} \Big] \tag{A20}$$

$$= \mathbb{E}_p\Big[ -\log \hat{p}_\theta(X_T) + \sum_{t=2}^T \log \frac{p(X_t|X_{t-1})}{\hat{p}_\theta(X_{t-1}|X_t)} + \log \frac{p(X_1|X_0)}{\hat{p}_\theta(X_0|X_1)} \Big] \tag{A21}$$

$$= \mathbb{E}_p\Big[ -\log \hat{p}_\theta(X_T) + \sum_{t=2}^T \log \Big( \frac{p(X_{t-1}|X_t, X_0)}{\hat{p}_\theta(X_{t-1}|X_t)} \cdot \frac{p(X_t|X_0)}{p(X_{t-1}|X_0)} \Big) + \log \frac{p(X_1|X_0)}{\hat{p}_\theta(X_0|X_1)} \Big] \tag{A22}$$

$$= \mathbb{E}_p\Big[ -\log \hat{p}_\theta(X_T) + \sum_{t=2}^T \log \frac{p(X_{t-1}|X_t, X_0)}{\hat{p}_\theta(X_{t-1}|X_t)} + \sum_{t=2}^T \log \frac{p(X_t|X_0)}{p(X_{t-1}|X_0)} + \log \frac{p(X_1|X_0)}{\hat{p}_\theta(X_0|X_1)} \Big] \tag{A23}$$

$$= \mathbb{E}_p\Big[ -\log \hat{p}_\theta(X_T) + \sum_{t=2}^T \log \frac{p(X_{t-1}|X_t, X_0)}{\hat{p}_\theta(X_{t-1}|X_t)} + \log \frac{p(X_T|X_0)}{p(X_1|X_0)} + \log \frac{p(X_1|X_0)}{\hat{p}_\theta(X_0|X_1)} \Big] \tag{A24}$$

$$= \mathbb{E}_p\Big[ \log \frac{p(X_T|X_0)}{\hat{p}_\theta(X_T)} + \sum_{t=2}^T \log \frac{p(X_{t-1}|X_t, X_0)}{\hat{p}_\theta(X_{t-1}|X_t)} - \log \hat{p}_\theta(X_0|X_1) \Big] \tag{A25}$$

$$= \mathbb{E}_p[\underbrace{D_{\text{KL}}(p(X_T|X_0) \parallel \hat{p}_\theta(X_T))}_{L_T}$$

$$+ \sum_{t=2}^T \underbrace{D_{\text{KL}}(p(X_{t-1}|X_t, X_0) \parallel \hat{p}_\theta(X_{t-1}|X_t))}_{L_{t-1}}$$

$$\underbrace{- \log \hat{p}_\theta(X_0|X_1)}_{L_0}] \tag{A26}$$

From this point on, it suffices with optimizing the intermediate term $L_{t-1}$ for all values of $t$. Note that we have an analytical expression for $p(X_{t-1} \mid X_t, X_0)$ which has a Gaussian form. Given that our model $\hat{p}_\theta(X_{t-1} \mid X_t)$ also parametrizes a Gaussian, we can exploit the closed-form solution for the KL divergence between Gaussian distributions, and leverage a simple change of variables (described in Ho et al. (2020)) in order to optimize a simpler expression:

$$L_{\text{simple}} = := E_{t\sim[1,T], X_0\sim p(X_0), \epsilon_t)}[||\epsilon_t - \epsilon_\theta(X_t, t)||^2] \tag{A27}$$

Here $\epsilon_t$ is the added noise at time $t$, and we train a model $\epsilon_\theta(X_t, t)$ to predict $\epsilon_t$ from Equation A7.

$L_{\text{simple}}$ does not provide any learning signal for $\Sigma_\theta(X_t, t)$. This happens because instead of learning $\Sigma_\theta(X_t, t)$, it is fixed to a constant $\beta_t\mathbf{I}$ as proposed by Ho et al. (2020).

$\mu_\theta(X_t, t)$ can be derived from $\epsilon_\theta(X_t, t)$ as follows:

$$\mu_\theta(X_t, t) = \frac{1}{\sqrt{\alpha_t}}\left( X_t - \frac{1-\alpha_t}{\sqrt{1-\bar{\alpha}_t}}\epsilon_\theta(X_t, t) \right) \tag{A28}$$

This sustitution is used both to derive Equation A27 and to sample. The sampling algorithm can be found in Ho et al. (2020). It is not the only possible sampler that has been proposed, but we use it for our method.

## A.3 EXPERIMENTAL DETAILS

In this section we describe the details for different aspects of the experiments: the overall setting and architecture, the datasets, details for LAION-400M, the sampling procedure and training regularizations.

**Architecture and Setting Details.** We design different architectures for different query-set types. Tables 4, 5 depict the designs for the *Querier*, and Table 6 for the *QuerySet Answerer*. For different datasets, The blocks may have different depths. For the patch-based querier, the number of channels is the sum of channels for the ground-truth reference image (if any), channels of the history and an extra channel that is a binary mask indicating what regions of the image are visible $C = C_{gt} + C_S + C_M$. The input is thus the naive concatenation of these three sources. For attribute-based query-sets, the input data is formatted differently. We have a two-channel one-dimensional vector that is the concatenation of an indicator of the answers to the visible attributes with values $\{-1, 0, 1\}$ and a binary mask indicating visibility.

We use Imagen Saharia et al. (2022) as the main generative pipeline. We slightly modify the conditioning methodology to adapt to our requirements. We describe the main modifications:

- Remove LayerNorm from fully connected layers that process attribute-like conditions.

- Attribute query-sets condition the U-Net through masked cross-attention in all layers, annulling the effect of queries that have yet to be asked. Diffusion time $t$ interacts with the U-Net solely through feature-wise modulation.

- Image-like conditions are concatenated to the noisy input image of the U-Net.

Next, we describe the main hyperparameters used for Imagen's U-Net. Learning rate: $LR = 1e - 4$ with a cosine decay; Base dimension: 32; Dimensionality multiplyers: $(1, 2, 4, 8)$, Self-attention at resolutions: $(\times \frac{1}{4}, \times \frac{1}{8})$; Query embedding size: $16 \times 2$; Condition size: 256; Number of steps for training and sampling: 256; Condition drop probability: $p = 0.1$. The reference code-base and other details can be found in `https://github.com/lucidrains/imagen-pytorch`. More details can be found in our codebase.

Table 4: Architecture for the querier used for the patch-based experiments.

| ConvBlock (C $\rightarrow$ 512) |
| --- |
| DeConvBlock (512 $\rightarrow$ 32) |
| Dense (32 $\rightarrow$ 1): Attention logits Hard Softmax: Selected Query |

Table 5: Architecture for the querier used for the attribute-based experiments.

| LinearBlock (|Q| $\rightarrow$ 2000) |
| --- |
| LinearBlock (2000 $\rightarrow$ 500) |
| Channel-concat Dense (1000 $\rightarrow$ |Q|): Attention logits Hard Softmax: Selected Query |

Table 6: Architecture for classifier used for the attribute-based experiments.

| LinearBlock (|Q| $\rightarrow$ 2000) |
| --- |
| LinearBlock (2000 $\rightarrow$ 500) |
| Channel-concat Dense (1000 $\rightarrow$ |Q|): Attribute logits |

Our method for LSUN Bedroom experiments has been trained as a wraper to Stable Diffusion V1-4: huggingface.co/CompVis/stable-diffusion-v1-4. We use the same version for the results displayed in Fig. 2.

When showing results for Stable Diffusion XL, we use the model in huggingface.co/stabilityai/stable-diffusion-xl-base-1.0.

**Quantitative experiments for the generative approach compositionality capability.** We choose as baselines the Stable Diffusion V1-4 baseline text-to-image model and Structured Diffusion Feng et al. (2023), a method to improve multi-concept conditioning in text-to-image models, specifically more accurate attribute binding and better image compositions. Both methods are conditioned by text, and thus we provide the attributes in the following format:

*"Photo of a bedroom with curtains that are light-colored, with white walls, with floor made of wood, with visible door, without an ensuite bathroom, with white bedsheets, and without a watermark."*

Note that we select 10 attributes from the query set of length 58. They correspond to the attributes with top entropy in the training dataset.

The reason why we do not provide the full list of attributes for the LSUN Bedroom dataset is that the baselines only accept prompts with a maximum length of 77 tokens, and thus we cannot concatenate all attributes in natural language.

**Datasets details.** Here, we describe the dataset details.

- **MNIST**: Training corpus consists of 60k $1 \times 32 \times 32$ greyscale images of handwritten single digits. In inference time we generate 5 samples for each example of the test set, consisting of 120 images. We generate samples for multiple description lengths, in order to generate the curves in Fig. 9. The test setting is the same for all experiments, except indicated otherwise.

- **CelebA**: It consists of 50k $3 \times 64 \times 64$ images of celebrity faces, divided into 34-1k-15k for training, validation and testing. The publicly released dataset can be found in https://www.tensorflow.org/datasets/catalog/celeb_a.

- **Clevr**: It consists of 8k (partitioned as 7k-1k-1k for training), validation and test. $3 \times 128 \times 128$ images generated by randomly placing objects in a flat background with different lightings. The objects have different controllable discrete attributes such as *color*, *shape* and *material*, as well as continuous attributes such as *rotation* or *size*. We choose color for our attribute-based experiments.

- **LSUN Bedroom and Churches with Queries**: Given the lack of existing datasets that met our task's requirements - adequate number of samples, high image quality, binary attribute descriptions and images belonging to a concrete and well-defined distribution, we created our own dataset. To do so, we selected a set of descriptive binary queries, some of which redundant with others, to categorize a certain image distribution. Bedroom dataset consists of 316k images, filtered to 60k and resized to $3 \times 512 \times 512$ belonging to LSUN (Large-scale Scene Understanding) dataset Yu et al. (2016) under the category 'bedroom'. Churches dataset consists of 70k images, filtered to 11k and split as $90\% - 10\%$, for training and validation, reserving 2k images for test. They belong to category 'churches' of LSUN and are also resized to $3 \times 512 \times 512$. Link to datasets provided in https://github.com/ArmandCom/InCoDe.

  Filtering includes discarding images with width/length ratios exceeding 1.3 or falling below 0.7, with additional rule-based filtering. The images were then passed through BLIP Li et al. (2022), a visual question-answering engine with a set of queries designed to describe the scene with binary answers. The official LSUN dataset can be found in: https://www.tensorflow.org/datasets/catalog/lsun.

  The list of binary attributes were the following (asterisk marks some level of redundancy):

  **Bedroom:**
    - Presence of curtains
    - Presence of red curtains*
    - Presence of brown curtains*
    - Presence of blue curtains*
    - Presence of light colored curtains*
    - Presence of a luggage
    - Presence of at least one person
    - Presence of an adult*
    - Presence of a child*
    - Presence of two children*
    - Presence of a blue wall
    - Presence of a red wall
    - Presence of a white wall
    - Presence of a dark wall
    - Presence of a window
    - Presence of sunlight coming out from a window*
    - Presence of plants visible from a window*
    - Presence of city buildings visible from a window*

- Visible floor
- Presence of a carpet*
- Wooden floor visible*
- Presence of a door
- Presence of an open door*
- Presence of door to an en-suite bathroom*
- Visible ceiling
- Presence of hanging lights from the visible ceiling*
- Presence of more than one bed
- Whether all beds are from the same size*
- Presence of bunk beds
- Whether the bed is big enough for two people
- Presence of a mosquito net
- Presence of a TV monitor
- Presence of a radio
- Presence of a radiator
- Presence of a bedside table
- Presence of a nightstand light in the bedside table*
- Presence of more than one bedside tables*
- Presence of photo frames in the bedside table*
- Presence of white bedsheets
- Presence of dark bedsheets
- Presence of blue bedsheets
- Presence of red bedsheets
- Presence of green bedsheets
- Whether the image is a collage
- Presence of an animal
- Presence of a dog*
- Presence of a cat*
- Presence of a bird*
- Presence of a hunting trophy in a wall*
- Presence of a telephone
- Presence of plants inside the room
- Whether the image is from a hotel room
- Presence of paintings in a wall
- Whether the lights are on
- Whether the bed is made
- Presence of a closet
- Presence of a wooden closet*
- Presence of an image watermark

**Churches:**

- Building made of stone
- If building is made of stone, is it a light-colored stone*
- Presence ofgreen grass in the image
- Presence of trees in the image
- If trees in the image, are they large trees*
- Presence of person in the image
- If person appears in the image, are they standing near the building*
- Presence of a chimney on the roof of the building
- Presence of windows in the building

- If building has windows, are they large windows*
- Wall of the building made of visible bricks
- If wall of the building is made of visible bricks, are the bricks red*
- Sky clear
- If sky is clear, is the sun visible in the image*
- Clouds in the sky
- Presence of more than one building in the image
- Is asphalt visible in the image
- Does the image have a watermark
- Does the building have a bell on the roof
- Presence of garden in the building
- Building in a rural area
- Presence of a mountain in the image
- Nighttime
- Building in the image from the Romanesque architectural style
- Building in the image from the Gothic architectural style
- Building in the image from the Renaissance architectural style
- Building in the image from the Baroque architectural style
- Building located near a body of water
- Presence of a path leading to the building
- Presence of a bell tower in the building
- Presence of a bench near the building
- Building surrounded by a fence
- Building made of wood
- Presence of columns in the building
- Presence of a large rose window in the building
- Does the building have a cross on the roof
- Presence of a bell tower separate from the main building
- Building adorned with sculptures
- If building is adorned with sculptures, are they religious figures*
- Presence of arches or vaulted ceilings inside the building
- Roof of the building tiled
- If the roof is tiled, are the tiles red*
- Presence of a statue of a saint or angel on the building
- Building painted

**Image Sampling.** Image sampling is done with the same algorithm as described in Saharia et al. (2022), the main difference being that the diffusion model is conditioned on the selected description $D$.

**Hardware** `InCoDe` has been trained in two NVIDIA GeForce RTX 2080 Ti GPUs. For images of resolution $64 \times 64$, it takes $\sim 1$ day to train. There are slight fluctuations due to model variations. The binary attribute image classifier has been trained in two NVIDIA RTX A6000 GPUs during $\sim 3$ days.

## A.4 CONDITIONAL DIFFUSION FROM THE PERSPECTIVE OF INFORMATION THEORY: PROOF OF PROPOSITION 2.1

**Initial remarks.** We wish to approximate $p(X_0)$ with our generative model $\hat{p}(X_0)$. Next, we provide derivation of the diffusion objective by means of the variational lower bound. Steps are

skipped for brevity. An expansion can be found in Ho et al. (2020) and Section A.2.2 of this appendix.

$$
\begin{aligned}
\mathbb{E}_{p(X_0)}[-\log \hat{p}(X_0)] &\leq \mathbb{E}_{p(X_0)}[-\log \hat{p}(X_0) + D_{\mathrm{KL}}(p(X_{1:T} \mid X_0) \,||\, \hat{p}(X_{1:T} \mid X_0))] \\
&= \mathbb{E}_{p(X_{0:T})}\left[\log \frac{p(X_{1:T} \mid X_0)}{\hat{p}(X_{0:T})}\right] \\
&= \mathbb{E}_{p(X_{0:T})}\big[D_{\mathrm{KL}}(p(X_T \mid X_0) \,||\, \hat{p}(X_T)) \\
&\quad + \sum_{t=2}^{T} D_{\mathrm{KL}}(p(X_{t-1} \mid X_0, X_t) \,||\, \hat{p}(X_{t-1} \mid X_t)) \\
&\quad - \log \hat{p}(X_0 \mid X_1)\big]
\end{aligned}
\tag{A29}
$$

Effectively we optimize the following objective, which is simplified.

$$
\begin{aligned}
&\min_{\hat{p} \in \mathcal{P}_{\mathcal{X}}} \mathbb{E}_{p(X_0), t \in [1,T]}[D_{\mathrm{KL}}(p(X_{t-1} \mid X_0, X_t) \,||\, \hat{p}(X_{t-1} \mid X_t))] + C \\
&\equiv \min_{\hat{p} \in \mathcal{P}_{\mathcal{X}}} \mathbb{E}_{X_0, \epsilon, t \sim [1,T]}\left[\|\epsilon_t - \epsilon_\theta(X_t, t)\|^2\right] = L_{\mathrm{simple}},
\end{aligned}
\tag{A30}
$$

where $C$ is a constant not depending on the parameters $\theta$ of $\hat{p}$ and $\epsilon_\theta(X_t, t)$ as our noise estimation model. Instead of a summation we take the expectation for $t \in [1, T]$. Minimizing $L_{\mathrm{simple}}$ is empirically shown to also minimize the KL divergence term in Equation A29 (Ho et al. (2020)). Here, we are interested in generating $X$ conditioned to a set of conditions $c \in \{c_1, \ldots, c_N\}$. Ho & Salimans (2021) makes the following connection:

$$
\begin{aligned}
&\min_{\hat{p} \in \mathcal{P}_{\mathcal{X}}} \mathbb{E}_{p(X_0 | c)}[\hat{p}(X_0 \mid c)] \\
&\leq \min_{\hat{p} \in \mathcal{P}_{\mathcal{X}}} \mathbb{E}_{p(X_0 | c), t \in [1,T]}[D_{\mathrm{KL}}(p(X_{t-1}) \mid X_0, X_t, c) \,||\, \hat{p}(X_{t-1} \mid X_t, c)] + C \\
&\equiv \min_{\hat{p} \in \mathcal{P}_{\mathcal{X}}} \mathbb{E}_{p(X_0 | c), \epsilon, t \sim [1,T]}\left[\|\epsilon_t - \epsilon_\theta(X_t, t, c)\|^2\right]
\end{aligned}
\tag{A31}
$$

Ho et al. (2020) only considers the unconditional distribution. The above is true if we consider $p(X_0 \mid c)$ as being a separate distribution, and our model $\hat{p}(X_0 \mid c)$ equivalent to having a family of unconditional models with different $\theta$ for each condition $c$. Therefore: $p(X_0 \mid c) \equiv p_c(X_0)$ and $\hat{p}_\theta(X_0 \mid c) \equiv \hat{p}_{\theta,c}(X_0)$.

Next, we prove the following lemma:

**Lemma A.2.** *Let $Q$ be a user-defined query set and $\mathbb{P}(X_t)$ all possible distributions on $X_t$ with $t$ being an arbitrary noise level of the diffusion process. Then, for any realization $S = s$, the following holds true:*

$$
\min_{\hat{p} \in \mathcal{P}_{\mathcal{X}}, q \in Q} \mathbb{E}_{p(X_0 | X_t, s)} \sum_{t=2}^{T} [D_{\mathrm{KL}}(p(X_{t-1} \mid X_0, X_t, q(X_0), s) \,||\, \hat{p}(X_{t-1} \mid X_t, q(X_0), s)]
$$
$$
\equiv \max_{\hat{p} \in \mathcal{P}_{\mathcal{X}}, q \in Q} I(q(X_0); X_1 \mid s)
\tag{A32}
$$
$$
- \mathbb{E}_{p(X_0 | X_t, s)} \sum_{t=2}^{T} [D_{\mathrm{KL}}(p(X_{t-1} \mid X_t, q(X_0), s) \,||\, \hat{p}(X_{t-1} \mid X_t, q(X_0), s))]
\tag{A33}
$$

**Proof of Lemma A.2:** With $c = \{q(X_0), s\}$, we express our objective in Equation (A1) in terms of mutual information.

$$\min_{\hat{p} \in \mathcal{P}_\mathcal{X}, q \in Q} \mathbb{E}_{p(X_0|X_t,s)} \sum_{t=2}^{T} [D_{\mathrm{KL}}(p(X_{t-1} \mid X_0, X_t, q(X_0), s) \| \hat{p}(X_{t-1} \mid X_t, q(X_0), s)]$$

$$= \min_{\hat{p} \in \mathcal{P}_\mathcal{X}, q \in Q} \mathbb{E}_{p(X_0|X_t,s)} \sum_{t=2}^{T} \left[ \sum_{X_{t-1}} p(X_{t-1} \mid X_0, X_t, q(X_0), s) \log \frac{p(X_{t-1} \mid X_0, X_t, q(X_0), s)}{\hat{p}(X_{t-1} \mid X_t, q(X_0), s)} \right]$$

$$= \min_{\hat{p} \in \mathcal{P}_\mathcal{X}, q \in Q} \mathbb{E}_{p(X_0|X_t,s)} \sum_{t=2}^{T} \left[ \sum_{X_{t-1}} p(X_{t-1} \mid X_0, X_t, q(X_0), s) \log \frac{p(X_0, X_{t-1} \mid X_t, q(X_0), s)}{\hat{p}(X_{t-1} \mid X_t, q(X_0), s) p(X_0 \mid X_t, q(X_0), s)} \right]$$

$$= \min_{\hat{p} \in \mathcal{P}_\mathcal{X}, q \in Q} \mathbb{E}_{p(X_0, X_{t-1}|X_t,s)} \sum_{t=2}^{T} \left[ \log \frac{p(X_0, X_{t-1} \mid X_t, q(X_0), s)}{p(X_{t-1} \mid X_t, q(X_0), s) p(X_0 \mid X_t, q(X_0), s)} \right]$$

$$+ \mathbb{E}_{p(X_0, X_{t-1}|X_t,s)} \sum_{t=2}^{T} \left[ \log \frac{p(X_{t-1} \mid X_t, q(X_0), s)}{\hat{p}(X_{t-1} \mid X_t, q(X_0), s)} \right]$$

$$= \min_{\hat{p} \in \mathcal{P}_\mathcal{X}, q \in Q} \sum_{t=2}^{T} [I(X_0; X_{t-1} \mid X_t, q(X_0), s)]$$

$$+ \mathbb{E}_{p(X_0|X_t,s)} \sum_{t=2}^{T} [D_{\mathrm{KL}}(p(X_{t-1} \mid X_t, q(X_0), s) \| \hat{p}(X_{t-1} \mid X_t, q(X_0), s))]$$

$$\tag{A34}$$

with $X = X_0$. Now observe that for any fixed $S = s$ and any $q \in Q$,

$$I(X_0, q(X_0); X_{t-1} \mid X_t, s) = I(X_0; X_{t-1} \mid X_t, s) + I(q(X_0); X_{t-1} \mid X_0, X_t, s)$$
$$= I(X_0; X_{t-1} \mid X_t, s) \tag{A35}$$

Decomposing $I(X_0, q(X_0); X_{t-1} \mid X_t, s)$,

$$I(X_0, q(X_0); X_{t-1} \mid X_t, s) = I(q(X_0); X_{t-1} \mid X_t, s) + I(X; X_{t-1} \mid X_t, q(X_0), s) \tag{A36}$$

We find that:

$$\min_{q \in Q} I(X_{t-1}; X_0 \mid X_t, q(X_0), s) \equiv \min_{q \in Q} -I(q(X_0); X_{t-1} \mid X_t, s). \tag{A37}$$

Then, we apply the equality in A37 to the first term of the last expression in Equation in A34:

$$\sum_{t=2}^{T} I(q(X_0); X_{t-1} \mid X_t, s) = \sum_{t=2}^{T} I(q(X_0); X_{t-1} \mid \{X_t, \dots, X_T\}, s) =$$
$$= I(\{X_1, \dots, X_T\}; q(X_0) \mid s) = I(X_1; q(X_0) \mid s) \tag{A38}$$

We easily see that Lemma A.2 is proven by applying the resulting expression in the mutual information term of Equation A34.

**Proof of Proposition 2.1** Once again, we restate the proposed objective in Equation A1 (ommitting the parameters subscript for simplicity):

$$\min_{\theta, \psi} \quad \mathbb{E}_{t \sim [1,T], p(X_0|S)} D_{\mathrm{KL}} \left( p(X_{t-1} \mid X_t, X_0) \| \hat{p}(X_{t-1} \mid X_t, q(X_0), S) \right) \tag{A39}$$
$$\text{where} \quad q = g(S) \in Q$$

We then proceed as follows:

$$\min_{\theta,\psi} \mathbb{E}_{p(X_0|s)} \sum_{t=2}^{T} \left[ D_{\text{KL}} \left( p(X_{t-1} \mid X_t, X_0) \mid\mid \hat{p}(X_{t-1} \mid X_t, q(X_0), s) \right) \right] \tag{A40}$$

$$= \mathbb{E}_{p(X_0|s)} \sum_{t=2}^{T} \left[ D_{\text{KL}} \left( p(X_{t-1} \mid X_t, X_0) \mid\mid \hat{p}_s^*(X_{t-1} \mid X_t, q_s^*(X_0), s) \right) \right]$$

$$= \mathbb{E}_{p(X_0|s)} \sum_{t=2}^{T} \left[ D_{\text{KL}} \left( p(X_{t-1} \mid X_t, X_0) \mid\mid \hat{p}(X_{t-1} \mid X_t, \hat{q}(X_0), s) \right) \right]$$

$$+ \mathbb{E}_{p(X_0|s)} \sum_{t=2}^{T} \left[ \sum_{X_{t-1}} p(X_{t-1} \mid X_t, X_0) \log \frac{\hat{p}(X_{t-1} \mid X_t, \hat{q}(X_0), s)}{\hat{p}_s^*(X_{t-1} \mid X_t, q_s^*(X_0), s)} \right]$$

$$= \mathbb{E}_{p(X_0|s)} \sum_{t=2}^{T} \left[ D_{\text{KL}} \left( p(X_{t-1} \mid X_t, X_0) \mid\mid \hat{p}(X_{t-1} \mid X_t, \hat{q}(X_0), s) \right) \right]$$

$$- \mathbb{E}_{p(X_0|s)} \sum_{t=2}^{T} \left[ \sum_{X_{t-1}} p(X_{t-1} \mid X_t, X_0) \log \frac{\hat{p}_s^*(X_{t-1} \mid X_t, q_s^*(X_0), s)}{\hat{p}(X_{t-1} \mid X_t, \hat{q}(X_0), s)} \right]$$

$$= \mathbb{E}_{p(X_0|s)} \sum_{t=2}^{T} \left[ D_{\text{KL}} \left( p(X_{t-1} \mid X_t, X_0) \mid\mid \hat{p}(X_{t-1} \mid X_t, \hat{q}(X_0), s) \right) \right]$$

$$- \mathbb{E}_{p(X_0|s)} \sum_{t=2}^{T} \left[ D_{\text{KL}} \left( \hat{p}_s^*(X_{t-1} \mid X_t, q_s^*(X_0), s) \mid\mid \hat{p}(X_{t-1} \mid X_t, \hat{q}(X_0), s) \right) \right]$$

$$\leq \mathbb{E}_{p(X_0|s)} \sum_{t=2}^{T} \left[ D_{\text{KL}} \left( p(X_{t-1} \mid X_t, X_0) \mid\mid \hat{p}(X_{t-1} \mid X_t, \hat{q}(X_0), s) \right) \right], \tag{A41}$$

for any realization $S = s$ with $p(S = s) > 0$. The optimal denoiser $\hat{p}_s^*$ and query $q_s^*$ are the solution to the minimzation problem. We denote any denoiser as $\hat{p}_s$, and $\hat{q} = g(s)$ is the output of any querier. We make use of Lemma A.2 in the fourth equality. We appeal to the KL divergence non-negativity for the inequality. This inequality holds for $\forall S = s$. We conclude that $q_s^* = g^*(s)$ and $\hat{p}_s^* = \hat{p}^*(X_t, \{q_s^*, q_s^*(x_0)\} \cup s)$ for any given $s$. Note than in a slight abuse of notation we denote both the denoising mapping and the posterior probability as $\hat{p}$. The Theorem A2 is proved by using Lemma A.2 to characterize $q_s^*$ and $\hat{p}_s^*$.

Note that parts of this proof are structurally equivalent to that of Chattopadhyay et al. (2023). However, we apply it to our particular case of image generation by denoising.

**Training Objective Function.**  Finally, applying the empirical equivalency in A30, we can substitute the KL divergence term in our objective function, obtaining the following expression:

$$\min_{\theta,\phi} \quad \mathbb{E}_{t\sim[1,T],X_0|S,\epsilon_t} \left[ \|\epsilon_t - \epsilon_\theta(X_t, t, S \cup \{q, q(X_0)\})\|^2 \right] \tag{A42}$$
$$\text{where} \quad q = g_\phi(S) \in Q$$

with $\epsilon_\theta$ as the noise estimation model with parameters $\theta$ and $g_\phi$ as the querier, with trainable parameters $\phi$.

## A.5  IMPACT

This paper presents work whose goal is to advance the field of Machine Learning and provide useful tools for artists to create and edit image content. The ethics of creating realistic images involve a complex interplay of factors such as intent, consent, impact, and cultural sensitivity. While realistic images can serve legitimate purposes, responsible creation and use, transparency, and respect for individuals' rights are essential to navigate the ethical considerations involved.

