# OpenReview forum: "InCoDe: Interpretable Compressed Descriptions For Image Generation"
_ICLR.cc/2025/Conference — ICLR 2025 Poster_

### Official Review · Reviewer_DRti · 2024-10-30

**Soundness:** 2
**Presentation:** 2
**Contribution:** 3
**Rating:** 6
**Confidence:** 2

**Summary:**

To improve generative modelling for multi-concept inputs, the paper proposes the Interpretable 'Compressed Descriptions for Image Generation' (InCoDe) Framework, which consists of a query encoder, an answer decoder, and a diffusion model.

The proposed framework sidesteps natural language to condition the generator. If I understand correctly, concept vectors replace natural language embeddings. The concept vector is constructed using a visual question answerer in tandem with an Embedder + MLP combination. The resulting embedding is fed into a pre-trained diffusion model.

The methods section builds upon the work of Chattopadhyay et al. The paper's key innovation is that we can modify the answers from the visual question answerer, which leads to changed images.

The proposed framework is experimentally tested on the CLEVER, LSUN Bedroom, and LSUN Churches Datasets.

**Strengths:**

The work focuses on interpretability and is relevant to the community. The paper presents a solid extension of Chattopadhyay et al.'s (2023) work in an image generation setting.

**Weaknesses:**

Evaluation:

It would have been nice to compare it to an off-the-shelf diffusion model (https://proceedings.neurips.cc/paper/2020/file/4c5bcfec8584af0d967f1ab10179ca4b-Paper.pdf) conditioned on, for example, the facial attributes of the CelebA dataset.
The paper's introduction claims that generative models struggle when asked to compose images with multiple concepts. While Figure 2 illustrates the situation qualitatively, it would have been nice if the paper had followed up with a quantitative analysis of the situation, including a comparison to established work. Table 2 might be doing this. I am not sure if lines 342 to 351 refer to Table 2. Please clarify in the rebuttal.


Minor points:

- Links to the supplementary material are broken. Please consider submitting single files in the future. The author guide specifically encourages single file submission ( https://iclr.cc/Conferences/2025/AuthorGuide ).
 -  The writing was hard to follow at times. Consider lines 342 to 351, for example. It would have been nice to mention and link to a table where readers can find the numerical results right there.

**Questions:**

Figure 3:

- Why does the QueryAnswerer in Figure 3 compute $\hat{Q}(X)$ instead of $\hat{Q}({\hat{X}})$? Arent we minimizing the a
cost term with $\hat{Q}({\hat{X}})$?

Equation two:

- How is the constraint in equation two enforced in practice?
- Do we need the decoder if we are mainly interested in the modified image?

Figure 8:

- What does DT-IC refer to?
- What does TopK-IC refer to?
- Are these related to lines 342 to 351?

Table 2:
- What do Stab. D and Stu. D refer to?
- Are these related to lines 342 to 351?

---

> ### Author Response · Authors · 2024-11-22
> **Response to Reviewer DRti**
>
> We thank reviewer DRti for their feedback. We leverage it to present new results. Below, we address their concerns and hope to have clarified their doubts. We are happy to provide further clarifications if needed.
>
> ## Evaluation
>
> > Comparison to stablished TTI [...] not sure if lines 342 to 351 refer to Table 2. Please clarify.
>
> Table 2 is indeed comparing our method to established work for text-to-image generation using diffusion models. Lines 342 to 351 describe the baselines used in this experiments and Section 3.2 (Composable generation paragraph) describes the results. We will add a link to the table in this paragraph to make it easier for the reader.
>
> We include a new baseline in our table. We compare to Accuracy and F1 scores of Stable Diffusion XL v1 (SD XL), one of the most advanced open models (note that DallE-3 in Fig. 2 is private and owned by OpenAI). Due to time constraints, we evaluate SD XL using 500 generated samples for both the Bedroom and Churches datasets. Our results show that even a model approximately 6.6 times larger than ours still struggles to combine multiple concepts and only slightly improves upon the results from its predecessor, SD V1. Results are the following:
>
>
> |          | Bedroom Acc | Bedroom F1 | Churches Acc | Churches F1  |
> | -------- | --------    | --------   | --------     | --------     |
> | SD V1    | 0.61        |0.62        | 0.57         | 0.55        |
> | SD XL    | 0.58        |0.64        | 0.59         | 0.61         |
> | InCoDe (SD V1)| **0.85**    |**0.84**    |**0.75**      |**0.72**|
>
> > Compare it to an off-the-shelf diffusion model [original DDPM paper] conditioned on facial attributes of the CelebA dataset.
>
> For CelebA with attributes, we followed the technique suggested by the reviewer in our initial submission. Specifically, we used an implementation of the Imagen model (a DDPM sampler) and trained it from scratch, conditioning on questions and answers based on CelebA attributes. Qualitative results can be found in Figure 18.
>
> The fine-tuning procedure shown in Figure 4 was applied only to the two LSUN datasets, which are the most challenging. While training from scratch is a feasible option, it may result in lower image quality and longer training times compared to the fine-tuning approach we propose.
>
> While we didn't have time to include this in the rebuttal, we propose, upon request, to compare CelebA trained from scratch with CelebA trained using our fine-tuning method in the final version of this work.
>
> ## Questions
>
> > $\hat{Q}(\hat{X})$ instead of $\hat{Q}(X)$?
>
> We believe $\hat{Q}({X})$ is adequate. Our Querier Answerer takes histories S (partial query-answer sets) and predicts all answers of queries applied to the **real** image X. Hence, there is never a generated sample $\hat{X}$ in this branch of the training pipeline.
>
> > Enforcing constraints in Eq. 2 in practice.
>
> These constraints are enforced by construction. The constraints indicate that 1) $q$ is the output of a querier function, modeled with our Querier and 2) the posterior is modeled by a function $f$, which corresponds to the Query Answerer.
>
> To aid clarity, we substitute "subject to" by "where", in Equation 2 (and equivalents).
>
> Please let us know if further clarification is needed.
>
> > Do we need a decoder if we are interested in the modified image?
>
> Our ultimate interest is to generate images that respect the semantics dictated by queries and their answers. This indeed allows us to modify generated images by changing answers to queries. However, the Querier and Query Answerer are employed for data representation rather than generation.
>
> The diffusion model is trained to generate images based on descriptions formed from query-answer pairs. The Querier's role is to select the most informative queries that create a concise, information-dense, query-based description. It is trained using the Decoder (Query Answerer), which predicts answers to all queries without requiring an image X as input, only histories S.
>
> Equation 2 outlines the objective function for training the Querier, where the image generator is not involved, as its training is decoupled from that of the Querier.
>
> > What are DT-IC and TopK-IC, are they related to L342-351?
>
> These baselines are entropy-based methods for selecting informative queries. They are described in more detail in lines 322-341. Please let us know if further clarification is needed.
>
> > What are Stab. D and Stru. D, are they related to L342-351?
>
> Stab. D (Stable Diffusion)(SD V1 in the revised version) and Stru. D (Structured Diffusion) are  text-to-image diffusion-based methods. Their description can indeed be found in lines 342-351. Again, let us know if further clarification is needed.

---

> ### Comment · Reviewer_DRti · 2024-11-28
>
> Thank you for responding, I will raise my score.

---

> > ### Author Response · Authors · 2024-11-28
> > **Thank you!**
> >
> > Thank you for reconsidering your score. We'd be happy to respond any remaining concerns.

---

### Official Review · Reviewer_vMKV · 2024-11-02

**Soundness:** 3
**Presentation:** 3
**Contribution:** 3
**Rating:** 6
**Confidence:** 3

**Summary:**

The paper proposes a framework called InCoDe, designed to improve the control and interpretability of image synthesis models. InCoDe leverages information-theoretic principles and compressed, interpretable descriptions, enabling users to generate images through a sequence of query-answer pairs. The model uses an Information Pursuit technique to select the most informative queries for defining images and then generates images using a diffusion model based on these descriptions. InCoDe’s primary modules include a Querier, Decoder, and Generator, which work together to iteratively refine image generation given user's attributes.

**Strengths:**

+ The paper is well written and the authors have done a good hob in providing the motivation for the paper. In addition, the given examples throughout the paper help providing intuition and a more clear understanding.

+ Information-Theoretic Foundation: The method relies on information-theoretic principles to determine query relevance and hence, prioritizes the most informative attributes.

+ Conditioning pre-existing diffusion models to harness their prior while "smoothing" the training procedure is interesting.

**Weaknesses:**

- The theoretical foundations are mainly taken from [1], and adapted into a new domain (text-to-image generation)

- Experimental part - the used models are relatively obsolete - the method is demonstrated on SD 1.4 while there are newer and stronger model, that perhaps more capable of following user-specified instructions. Moreover, utilizing BLIP as an evaluation does not express the generated images quality. Perhaps inCode is capable of following the user specified attributes better but while decreasing images quality? I believe that images quality evaluation (e.g., FID) should be considered.

- The method requires pre-defined set of attributes, which is feasible in simple datasets, but significantly less in real-world datasets such as ImageNet or COCO.

- Minor - error in the axis names in Fig 8?


[1] VARIATIONAL INFORMATION PURSUIT FOR INTERPRETABLE PREDICTIONS

**Questions:**

Please see weaknesses.

---

> ### Author Response · Authors · 2024-11-22
> **Response to Reviewer vMKV**
>
> We thank the reviewer for their both their comments. Their constructive feedback let us to perform experiments that will indeed make this work more complete. Below we address all their concerns.
>
> ## Theoretical foundations
>
> > Theoretical foundations are mainly taken from [1] and adapted.
>
> This is true. We do not claim to have introduced Variational Information Pursuit (V-IP), which is the contribution of [1]. Instead, in our work we present an adaptation of V-IP that helps us solve the problem at hand: selecting the most informative queries to achieve succint query-answer descriptions in the context of data representation for image generation.
>
> This paper presents algorithmic and experimental contributions, instead of theoretical ones. We propose a novel framework introducing new elements to achieve our objective, such as adapting the theoretical principles in [1] to the paradigm of generation or proposing a novel generative method.
>
> [1] Chattopadhyay, Aditya et al. “Variational Information Pursuit for Interpretable Predictions.” International Conference on Learning Representations (2023).
>
> ## Evaluation
>
> > Used models are Relatively obsolete [...] There are newer and stronger models, perhaps more capable.
>
> We agree with the reviewer that this is a relevant comparison. Therefore, we compare Accuracy and F1 scores with Stable Diffusion XL v1 (SD XL), one of the most advanced open models (note that DallE-3 in Fig. 2 is private and owned by OpenAI). Due to time constraints, we evaluate SD XL using 500 generated samples for both the Bedroom and Churches datasets. Our results show that even a model approximately 6.6 times larger than ours still struggles to combine multiple concepts and only slightly improves upon the results from its predecessor, SD V1. Results are as follows:
>
>
> |          | Bedroom Acc | Bedroom F1 | Churches Acc | Churches F1  |
> | -------- | --------    | --------   | --------     | --------     |
> | SD V1    | 0.61        |0.62        | 0.57         | 0.55        |
> | SD XL    | 0.58        |0.64        | 0.59         | 0.61         |
> | InCoDe (SD V1)| **0.85**    |**0.84**    |**0.75**      |**0.72**|
>
> > BLIP does not express generated images quality [...] FID should be considered.
>
> FID may not be an ideal quality metric for this task.
> Our base model and baseline, SD V1, has been trained with LAION-5B Dataset, of which LSUN data is just a small portion of approximately 0.02%.
>
> FID tends to be more informative when comparing similar distributions, but here, we are comparing images from LSUN (a specific subset) to a model trained on a broader distribution (LAION-5B).
>
> Therefore, we anticipate FID scores higher than usual.
>
> InCoDe, a wrapper for SD V1, was fine-tuned with a low learning rate to preserve the prior distribution learned by SD V1. As a result, we expect the FID to be improved, but not necessarily excellent. However, this does not necessarily mean that the generated images are not realistic.
>
> In our figures we show samples generated by different models (Figs. 1, 2, 5, 12, 13, 15, 16, 17) where good generation quality can be appreciated. We can provide more random samples of different models at the request of the reviewer.
>
> We conducted experiments by center-cropping the 2000 test images for both Bedroom and Churches datasets, and computing FID with our base model SD V1 and InCoDe. The results show significant improvements in the Churches dataset and modest gains in the Bedroom dataset:
>
> |     FID  | LSUN Bedroom | LSUN Churches|
> | -------- | --------    | --------   |
> | SD V1    | 57.13  | 89.23 |
> | InCoDe (SD V1)  |**55.14**  | **34.11** |
>
>
> FID was computed with this codebase: https://github.com/mseitzer/pytorch-fid. Results will be included in the final version of the paper at the request of the reviewer.
>
> ## Limitations
>
> > Method requires pre-defined set of attributes [...] less feasable in real-world datasets.
>
> While we would argue that Bedroom or Church datasets are real-world datasets, it is true that they have less variety in their semantic elements than Imagenet, for instance.
>
> In the case of widely variable datasets, if a user would want to capture certain tasks, this would require either larger query-sets, or richer answers to questions (non-binary). While we do not present experiments to that extent, there are no inherent limitations for our framework to accept these changes.
>
> However, as it is true for any generative model, to capture more semantic elements we would likely need to increase the model and dataset sizes, along with longer training times.
>
> ## Minor
> > Error in axis names Fig. 8?
>
> This is true, we correct the figure in the revised version.

---

> > ### Comment · Reviewer_vMKV · 2024-11-26
> > **Reviewer vMKV Rebuttal's response**
> >
> > I appreciate the authors response and the conducted experiments.
> >
> > **Evaluation**
> >
> > While 2000 samples are not a lot, the gaps from the baseline are significant, mitigating my concern regarding the potential degradation in image quality.
> >
> > **Limitations**
> >
> > My original concern was that in the experimental setting, the images are from a specific domain in which it is feasible to define sets of attributes. However, in the real-world scenario, how can the method be implemented effectively where there are practically infinite number of attributes? I view this as an inherent drawback of the method.

---

> > > ### Author Response · Authors · 2024-11-26
> > > **Response to reviewer vMKV**
> > >
> > > We thank the reviewer for acknowledging our improvement in evaluation.
> > >
> > > Regarding the limitations of our model, we’d like to point the reviewer to the Generalization section in General comments, where we discuss the trade-off between generality and specialization.
> > >
> > > As noted by reviewer DhcA, our method works well within its scope, which is to specialize to a well-defined image semantics space, rather than broad generalization.
> > >
> > > Our model is indeed limited in representing an infinite number of attributes. However, it is reasonable to assume that a potential user of a generative model might prioritize higher accuracy in capturing a self-defined, restricted set of attributes over having virtually unlimited flexibility at the cost of significantly lower accuracy (Fig. 2, Tab. 2).
> > >
> > > As such, our framework is not designed to generate images from distributions like Imagenet or COCO under arbitrary semantic guidance. Instead, it excels in targeted tasks such as room decoration, human face generation, fashion design, or product design, where its specialized approach is particularly effective.
> > >
> > > We believe that our contribution brings value to the comunity by addressing practical use cases where other mainstream models are shown to fail.

---

> > > > ### Comment · Reviewer_vMKV · 2024-11-30
> > > > **Response**
> > > >
> > > > I thank the author for their response.
> > > > While I still view above concerns as limitations, I accept the claims of the authors and have raised my score.

---

### Official Review · Reviewer_K8Cj · 2024-11-04

**Soundness:** 3
**Presentation:** 3
**Contribution:** 3
**Rating:** 6
**Confidence:** 3

**Summary:**

The paper introduces a framework InCoDe designed to improve the interpretability and control of image generation models. InCoDe provides a user-friendly, interpretable interface for custom image generation, enhancing both semantic control and user experience. The authors use the compressed, interpretable descriptions based on query-answer pairs to guide image generation, addressing limitations in current text-to-image models that struggle with composing multiple concepts accurately. And they also collected two new datasets along with sets of binary queries and answers about their content.

**Strengths:**

1. **A Novel User Friendly Interface:** InCoDe’s query-answer framework allows users to control image generation through a sequence of intuitive questions, making it more approachable and interactive than text-based inputs. This structure lets users directly specify attributes without needing technical prompts.
2. **Enhanced Interpretability and Control:** By using interpretable, information-rich query chains, InCoDe enables users to understand and shape each step of the image generation process, offering a clear view of how specific attributes impact the final image. This makes image manipulation more transparent and precise.
3. **Comprehensive Experimental Validation:** The framework’s effectiveness is demonstrated across diverse datasets and scenarios, with strong performance in generating accurate, attribute-aligned images.

**Weaknesses:**

1. **Limited Flexibility Compared to Text Input:** While the query-answer format improves control, it may restrict expressiveness compared to free-text descriptions. Users can only manipulate images within the bounds of available queries, which might not capture all desired nuances or novel concepts.

2. **Potential Challenges in Real-World Scenarios:** InCoDe has been tested on curated datasets, but real-world image generation tasks may introduce more variability and ambiguity. Handling this with a fixed query-answer system may be challenging and could limit the framework’s effectiveness in broader, unstructured contexts.

3. **Reliance on Visual Question Answering (VQA) Accuracy:** The framework depends on accurate VQA responses to generate coherent descriptions and images. Errors in VQA performance could affect the quality and alignment of generated images, especially in complex or nuanced queries.

**Questions:**

See the weaknesses.

---

> ### Author Response · Authors · 2024-11-22
> **Response to reviewer K8Cj**
>
> We thank the reviewer for their accurate observations. Below we address their concerns.
>
> ### Limitted Flexibility
>
> > While control is improved, it restricts expressiveness.
>
> This is accurate. This work prioritizes specialization over generalization. Fortunately, users have flexibility in designing queries, allowing them to capture virtually any desired semantics. We discuss this topic in more detail in the General Comments section, under Generalization.
>
> We are glad to extend this discussion at the reviewer's request, and/or to include this discussion in the main paper.
>
> ### Real-World Scenarios
>
> > real-world image generation tasks my introduce more variability and ambiguity [...] might be challenging [...] could limit framework's effectiveness.
>
> While we would argue that the Bedroom and Church datasets are real-world datasets, it is true that they have less variety in their semantic elements than Imagenet, for instance.
>
> In the case of widely variable datasets, if a user would want to capture certain tasks, it would require either larger query-sets, or richer answers to questions (non-binary). While we do not present experiments to that extent, there are no inherent limitations for our framework to accept these changes.
>
> However, as it is true for any generative model, to capture more semantic elements we would likely need to increase the model and dataset sizes, along with longer training times.
>
>
> ### Reliance on VQA
>
> > framework depends on accurate VQA responses [...] errors in VQA could affect performance.
>
> We acknowledge this limitation. The effect of errors in VQA models is indeed an interesting study for future work. Fortunately, off-the-shelf VQA models often offer reasonable performance and they are the best systematic way to provide answers we have identified for textual queries. But as indicated, they may introduce error. That said, we’d like to underscore the following:
>
> 1. VQA models are primarily employed to annotate unlabeled training data. By using VQA models to answer every query for each sample in our training set, we can work with images that lack annotations, giving our framework extensive flexibility across various image distributions.
> However, in datasets such as CelebA, where each image is already paired with manually annotated attributes, we can bypass VQA-related errors during training by using the provided annotations.
>
> 2. At test-time, a VQA model is **not** strictly necessary. For any single image, query responses can be manually provided by a user, based on either (i) their semantic preferences or (ii) their visual analysis of a reference image.
>
> 3. Queries are not always textual, so models like BLIP may not be needed in every case. For example, in the expriments shown in the supplementary material (Sec. A.1.1), location-based queries are answered by referencing a specific part of the image, eliminating the need for a VQA model.
>
> Finally, there has been interesting recent work on fine-tuning general-purpose VQA models to enable explainable decisions in image classification datasets such as ImageNet and Places365 [1]. These method could potentially be beneficial at improving the VQA capabilities of the framework introduced in this paper.
>
> [1] Chattopadhyay, Aditya et al. “Bootstrapping Variational Information Pursuit with Large Language and Vision Models for Interpretable Image Classification.” International Conference on Learning Representations (2024).

---

### Official Review · Reviewer_DhcA · 2024-11-05

**Soundness:** 4
**Presentation:** 4
**Contribution:** 3
**Rating:** 8
**Confidence:** 4

**Summary:**

The authors propose a new method for selecting human-interpretable semantic descriptions that allow for precise and reliable control of diffusion image generative models. The authors find these descriptions via the Information Pursuit optimization with neural networks, collect datasets to support and evaluate their approach, and quantitatively and qualitatively validate the effectiveness of InCoDe (the proposed method) on four different datasets. Specifically, these datasets consist of image and query-answer pairs for each image that describe its semantic content. Out of the four datasets, two are new and created by the authors, where existing LSUN Bedroom and LSUN Church images are given relevant query-answer pairs to describe the images.

**Strengths:**

I find the paper generally well-written and polished. As a result, it was easy to read and understand. For instance, the mathematical equations are clear and notions used in this paper are explained in the appendix. The diagrams in Fig. 3 and Fig. 4 clearly show the how InCoDE is trained.

Making image generation more controllable via succinct, human-interpretable descriptions is of great value to the image generation creative community.

InCoDe is compared against solid baselines (Figure 8, Figure 9), where it achieves significant improvements in image quality.

The authors promise the newly created datasets will "be released for public use", which is a big plus given the lack of similar datasets at the moment.

The authors do a good job explaining the model architecture, datasets, and the hardware setup in A.3 EXPERIMENTAL DETAILS, which I believe makes the results reproducible.

**Weaknesses:**

The related works section is somewhat oversimplified -- I think the authors could do a better job offering more context on the current state of interpretability and controllability in image diffusion models.

Despite the effectiveness the authors have discovered with using InCoDe on the four dataset which they tested, it is not clear whether InCoDe can generalize to more "free-form" types of controllability. All four datasets include a fixed set of 40-58 queries that are hand-selected and deemed relevant, but in the real world, users use image diffusion models to generate highly diverse images. It would seem that they are out of luck if these images do not comply with one of the pre-defined set of queries.

**Questions:**

*minor questions and concerns from the reviewer*
Some figures, e.g., Figures 8 and 9 appear to be raster rather than vector graphics? They look a bit blurry when I zoom in.

**Details Of Ethics Concerns:**

The paper enhances user controllability for the generation of realistic images, which can raise ethics concerns. To this end, the authors also release an ethics statement in the appendix:

A.5 BROADER IMPACT
This paper presents work whose goal is to advance the field of Machine Learning and provide useful
tools for artists to create and edit image content. The ethics of creating realistic images involve a
complex interplay of factors such as intent, consent, impact, and cultural sensitivity. While realistic
images can serve legitimate purposes, responsible creation and use, transparency, and respect for
individuals’ rights are essential to navigate the ethical considerations involved.

---

> ### Author Response · Authors · 2024-11-22
> **Response to reviewer DhcA**
>
> We sincerely thank reviewer DhcA for their positive and constructive feedback on this work. Below, we address their concerns.
>
> ### Related Work
>
> > Related works section is somewhat oversimplified [...] state of interpretability and controllability in image diffusion models.
>
> We appreciate the reviewer’s suggestion and have updated our manuscript accordingly. We have added more citations and connected each control domain to its interpretability form. Please let us know if you recommend further expansion or a different direction.
>
> ### Generalization
>
> > it is not clear whether InCoDe can generalize to more "free-form" types of controllability [...]  users use image diffusion models to generate highly diverse images. [...] out of luck if these images do not comply with one of the pre-defined set of queries.
>
> This is true. This work prioritizes specialization over generalization. Fortunately, users have flexibility in designing queries, allowing them to capture virtually any desired semantics.
>
> We discuss this further in the section on Generalization of General Comments.
>
> We are glad to extend this discussion at the reviewer's request, and/or to include this discussion in the main paper.

---

> > ### Comment · Reviewer_DhcA · 2024-11-25
> >
> > The authors have addressed my main questions. While other reviewers pointed out that the approach is not easily adaptable to arbitrary forms of image controllability, this approach works well within its scope, which is to specialize in a well-defined set of image semantics rather than broad generalization. I maintain that this is valuable to the image generation community. I appreciate the authors' efforts to present new experiments, especially the use of the new SD XL backbone. I wish to maintain my score.

---

> > > ### Author Response · Authors · 2024-11-28
> > > **Thank you!**
> > >
> > > Thank you for maintaining your score. We appreciate your comments.

---

### Author Response · Authors · 2024-11-22
**General Comments**

We thank the reviewers for their constructive comments and valuable feedback. They found our work to be well-written and polished (DhcA, vMKV), easy to understand (DhcA), and well-motivated with clear examples and explanations (vMKV). Reviewers DhcA and K8Cj appreciated the enhanced control and interpretability offered by our approach, noting its value to the image generation community. The novelty and user-friendliness of our interface were highlighted (K8Cj), alongside our method's capability to achieve significant improvements over solid baselines (DhcA). Comprehensive experimental validation and the release of new datasets for public use were also recognized (DhcA, K8Cj). Finally, reviewers recognized our focus on interpretability (DRti), reinforcing the relevance and significance of our work to the field.

Some concerns have also been raised. We address them below and upload a revised version of the paper, with changes in red and a single file with the main manuscript and the supplementary material.

## Generalization

Reviewers DhcA and K8Cj correctly assessed that our work suffers from generality in exchange for capturing accurately the desired semantics. This is discussed in L86-88 and L221-229.

In this work, we design our framework with a specific type of problem in mind, where interpretability and alignment across multiple concepts in generation take precedence over broad generalization. As is often noted, there's no free lunch: by fine-tuning our model on a particular subset of images relevant to the user, with specific queries, we "specialize" the model for a targeted user application, though this reduces its ability to represent other types of data.

Fortunately, there’s no inherent limitation on the types of semantics or images we can use to train the model.

We can draw an analogy to the pre-training and post-training phases in LLM training. The pre-trained diffusion model learns a wide distribution of general semantics and understands the relationships between text and images, but this stage is costly in terms of time, computation, and data. In contrast, the second phase (this work) is much more efficient. By using a more focused dataset, few trainable parameters and shorter training times, we fine-tune the model to capture a specific subdistribution, steering it to generate samples that are particularly well-suited to the task, while still leveraging the broad knowledge from the expensive pre-trained model.

---

### Meta-Review · Area_Chair_7fr9 · 2024-12-18

**Metareview:**

This paper proposed a method to select interpretable semantic descriptions for precise control of diffusion models.

For the strength of the paper, most reviewers like the user-friendly and interpretable interface for custom image generation, experimental validation on 4 data sets against solid baselines, and extra data set contributions.

For the weakness of the paper, as two reviewers pointed out, the whole paper is on specifically curated data sets like bedrooms etc, and hence generalization to more diverse data set or real world data remains unknown. Other concerns include inflexibility of the proposed method compared to text to prompt, reliance on VQA accuracy, experiment results on old generated models (SD 1.4), etc.

Overall, the topic is practically important and the proposed method does provide a user-friendly and interpretable system to customize image generation. The experiment results look promising, and the two data sets are a good contribution to the research community too. So I would recommend as "Accept as poster".

**Additional Comments On Reviewer Discussion:**

Two reviewers DhcA and K8Cj proposed the question of whether the proposed method can generalize to more diverse data sets. The authors argued that it is a design choice of scarifying generality in exchange for capturing accurately the desired semantics, and the framework does not have limitation, and hence different specific domain can be applied by different models. For a similar comment by reviewer K8Cj on lack of flexibility compared to text input, similar argument are discussed by the authors.

Reviewer vMKV asked why no recent generative models than SD 1.4 is used, which is addressed by authors after providing experiiment results on SDXL. Multiple questions about technical detail are addressed by providing more clarification.

---

### Decision · Program_Chairs · 2025-01-22

Accept (Poster)